# Production of aerosol containing ice nucleating particles (INPs) by fast growing phytoplankton

Daniel C. O. Thornton[1], Sarah D. Brooks[2], Elise K. Wilbourn[1], Jessica Mirrielees[2], Alyssa N. Alsante[1], Gerardo Gold-Bouchot[1], Andrew Whitesell[1,3], Kiana McFadden[2,4]

[1]Department of Oceanography, Texas A & M University, O & M Building, College Station, 77843, Texas, United States

[2]Department of Atmospheric Sciences, Texas A & M University, O & M Building, College Station, 77843, Texas, United States

10  [3]North Carolina State University, Raleigh, 27695, North Carolina, United States

[4]Jackson State University, Jackson, 39217, Mississippi, United States

*Correspondence to*: Daniel C. O. Thornton (dthornton@tamu.edu) and Sarah D. Brooks (sbrooks@tamu.edu)

15  **Abstract.** Sea spray aerosol contains ice nucleating particles (INPs), which affect the formation and properties of clouds. Here, we show that aerosols emitted from fast growing marine phytoplankton produce effective immersion INPs, which nucleate at temperatures significantly warmer than the atmospheric homogeneous freezing (-38.0 ℃) of pure water. Aerosol sampled over phytoplankton cultures grown in a marine aerosol reference tank (MART) induced nucleation and freezing at temperatures as high as -15.0 ℃ during exponential phytoplankton growth. This was observed in monospecific cultures representative of two major groups of phytoplankton: a cyanobacterium (*Synechococcus elongatus*) and a diatom (*Thalassiosira weissflogii*). Ice nucleation occurred at colder temperatures (-28.5 ℃ and below), which were not different from the freezing temperatures of procedural blanks, when the cultures were in the stationary or death phases of growth. Ice nucleation at warmer temperatures was associated with relatively high values of the maximum quantum yield of photosystem II ($\Phi_{PSII}$), an indicator of the physiological status of phytoplankton. High values of $\Phi_{PSII}$ indicate the presence of cells with efficient photochemistry and greater potential for photosynthesis. For comparison, field measurements in the North Atlantic Ocean showed that high net growth rates of natural phytoplankton assemblages were associated with marine aerosol that acted as effective immersion INPs at relatively warm temperatures. Data were collected over 4 days at a sampling station maintained in the same water mass as the water column stabilized after deep mixing by a storm. Phytoplankton biomass and net phytoplankton growth rate (0.56 day$^{-1}$) were greatest over the 24 h preceding the warmest mean ice nucleation temperature (-25.5 ℃). Collectively, our laboratory and field observations indicate that phytoplankton physiological status is a useful predictor of effective INPs, and more reliable than biomass or taxonomic affiliation. Ocean regions associated with fast phytoplankton growth, such as the North Atlantic during the annual spring bloom, may be significant sources of atmospheric INPs.

## 1 Introduction

Atmospheric aerosols significantly influence the Earth's radiative budget and climate through direct (scattering and absorption of light) and indirect effects (modifying cloud formation, properties and lifetimes) (Seinfeld et al., 2016). Clouds form through interactions between aerosol and water vapor, with cloud condensation nuclei (CCN) catalysing the formation of water droplets and ice nucleating particles (INP) catalysing the formation of ice crystals (Andrea and Rosenfeld, 2008; Hoose and Mohler et al., 2012; Hudson and Noble, 2021). Ice crystals are present in clouds at all latitudes; over the equatorial and high-latitude ocean, the

ice water path in clouds is greater than the liquid water path (Boucher et al., 2013). In the absence of INPs, homogenous freezing

of water in the atmosphere occurs at temperatures below -38 ºC (Kanji et al., 2017; Vali et al., 2015). INPs catalyse the freezing of water at warmer temperatures through several heterogeneous freezing modes: immersion mode in which the supercooled liquid water droplet surrounds the INP, contact mode in which freezing is initiated by an INP which has collided with the surface of a supercooled liquid water droplet, or deposition mode in which ice grows on an INP from water vapor (Kanji et al., 2017; Vali, 1985; Vali et al., 2015). A major limitation in our ability to understand and make accurate predictions about the climate system are

uncertainties associated with aerosol and cloud properties (Boucher et al., 2013). The latest generation of climate models, particularly through the Coupled Model Intercomparison project (CMIP6), indicate that the climate system is even more sensitive to cloud dynamics than previously thought (Palmer, 2020; Zelinka et al., 2020). Warming decreases cloud water content and coverage more than previously predicted, resulting in a positive feedback and more warming (Palmer, 2020; Schneider et al., 2019; Zelinka et al., 2020). These examples emphasize the role of clouds in climate uncertainty and our need to better understand cloud

microphysics, to enable clouds to be better parameterized in global climate models (GCMs).

The ocean covers 71 % of the Earth's surface and is a major source of atmospheric aerosol through the production of sea spray aerosol (SSA) (Lohmann et al., 2016). Containing a significant fraction of organic matter (O'Dowd et al., 2004; Gant and Meskhidze, 2013; Quinn et al., 2014), SSA is a chemical link between atmosphere and oceans, and plays a role in climate (Cochran et al., 2017). There is a need to not only characterize SSA in the atmosphere, but also the biogeochemical processes in the ocean

that affect the precursors of SSA (Brooks and Thornton, 2018; Mansour et al., 2020; Prather et al., 2013; Wilbourn et al., 2020; Quinn et al., 2019; Saliba et al., 2019, 2020; Sanchez et al., 2021; Twohy et al., 2021). Number concentrations of INP from marine sources are poorly constrained, but generally considered to be orders of magnitude lower than number concentrations of INP from terrestrial sources, particularly mineral dust (Beall et al. 2022; DeMott et al., 2016). Observations indicate that number concentrations of INP over the remote ocean are low (0.38 to 4.6 INP m$^{-3}$ of air over the Southern Ocean nucleating at -20 ºC)

(McCluskey et al. 2018). Nevertheless, marine sources of INP are important over the remote oceanic areas, such as the Southern Ocean, due to the absence of terrestrial INP and large surface area of the Earth covered by the ocean (Zhao et al., 2021). The properties of SSA have been coupled to phytoplankton blooms, indicating potential relationships between ecological processes in the epipelagic ocean, the formation of SSA, and the properties of the clouds that form on marine aerosol (O'Dowd et al., 2004; Quinn et al., 2019; Saliba et al., 2019; Mansour et al., 2020; Croft et al., 2021). Phytoplankton assemblage composition,

physiological status, and interactions with other organisms are all potentially important in determining whether the organic matter produced by phytoplankton produces SSA containing effective INPs (Wilbourn et al., 2020). O'Dowd et al. (2015) found that SSA was more enriched with organic matter during bloom collapse, which was hypothesized to be due to the release of large amounts of organic matter into the water by dying cells. Exudates from phytoplankton have been proposed as a source of INPs in the sea surface microlayer (SML) (Wilson et al., 2015; Irish et al., 2017). Of the carbon fixed by phytoplankton photosynthesis, 2 to 50 %

is released into the surrounding water as dissolved organic matter (DOM) (Thornton, 2014). Potentially, this DOM contains representatives from all the major groups of biological compounds (carbohydrates, proteins, lipids and nucleic acids) (Thornton, 2014). The amount of DOM released by phytoplankton increases when cells are stressed by environmental factors associated with bloom collapse, such as nutrient limitation (Thornton, 2002; 2014).

Diatoms (Alpert et al., 2011a; Knopf et al., 2010; Wilson et al., 2015), coccolithophores (Alpert et al., 2011b),

cyanobacteria (Wilbourn et al., 2020; Wolf et al., 2019), and chlorophytes (Alpert et al., 2011b) are all known sources of effective INP. There is insufficient data to determine if taxonomic affiliation and associated functional traits (e.g. chemical composition) determine whether phytoplankton act as INPs in greater quantities or warmer temperatures. It is known that changes in

physiological status affect resource allocation and therefore the chemical composition of phytoplankton (Klausmeier et al., 2004; Van Mooy et al., 2009; Dyhrman et al., 2012).

The objective of this work was to evaluate the link between growth and the physiology of representative phytoplankton with the properties of primary marine aerosols, focusing on factors that determine whether the aerosol are effective INPs. Growth experiments under controlled conditions were conducted with representative phytoplankton taxa in a marine aerosol reference tank (MART). Aerosol were collected onto aluminium foil substrates and their ability to act as INPs in the immersion mode was measured using our well-established ice microscope technique (Fornea et al., 2006). These data were compared with ice nucleation

measurements on *in situ* aerosol collected in the North Atlantic during the annual spring bloom.

## 2 Methods

### 2.1 Growth of phytoplankton in a marine aerosol reference tank (MART)

Identifying the method which produces the most realistic primary aerosols in a laboratory is a challenge previously addressed by a number of research groups (Fuentes et al., 2010a, 2010b; Prather et al., 2013; Sellegri et al., 2006; Stokes et al., 2013). Since

using a bubble plume from a plunging jet of water (Fuentes et al., 2010a; Sellegri et al., 2006) has been shown to be the best method, we used a Marine Aerosol Reference Tank (MART) featuring a plunging jet. The MART provided a controlled environment for growing phytoplankton in batch culture and a closed headspace where atmospheric conditions above the ocean were simulated. It is a 227 liter acrylic aquarium tank (52 cm wide x 122 cm long x 33 cm deep), closed with a neoprene gasket and sheet of 5 mm acrylic. The water within the tank was stirred using three magnetic stir bars (6 cm long, at a rate of approximately

180 rpm) to maintain the phytoplankton in suspension (Fig. 1). Illumination was provided by four 112 cm LED lights (Fluence Biengineering) simulating daylight. Photon flux density (PFD) was varied via a 1-10V input dimmer. PFD within the tank was measured using a Photosynthetically Active Radiation (PAR) sensor (LiCOR, Inc.). Cultures were grown at a PFD of 1 $\mu$mol m$^{-2}$ s$^{-1}$ on a 14 h light: 10 h dark cycle. Temperature on the outer surface of the tank and room air were monitored continuously using calibrated thermocouples in Labview (National Instruments), and maintained at 27 to 28 °C. Prior to

introduction to the MART, phytoplankton were grown in artificial seawater (ASW) (Harrison et al., 1980; Berges et al., 2001) supplemented with trace metals and vitamins from the L1 medium recipe of Guillard and Hargraves (1993). The ASW was made with high purity analytical grade salts. Nevertheless, the large mass of salts in artificial seawater represents a source of potential contamination for ice nucleation experiments. To reduce this possibility, sodium chloride was combusted for 6 h at 500 °C to remove organic contamination. This precaution was taken given the large amount of sodium chloride in the medium (21.19 g L$^{-1}$

). For other salts, notably the hydrated salts, combustion to remove potential organic contamination was not an option as it would have changed the composition and solubility of the salts. The tank was inoculated with 0.63 L of exponentially growing culture (1 % of total culture volume), which was added to the 63 L of medium in the tank. *Thalassiosira weissflogii* (CCMP 1051) was grown as a representative diatom, and in separate experiment, *Synechococcus elongatus* (CCMP 1379) was grown as a representative cyanobacterium. Macronutrient concentrations were 60 $\mu$M NaNO$_3$, 20 $\mu$M NaH$_2$PO$_4$, and 60 $\mu$M Na$_2$SiO$_3$ in the *T. weissflogii*

culture and 100 $\mu$M NaNO$_3$, 50 $\mu$M NaH$_2$PO$_4$, and 60 $\mu$M Na$_2$SiO$_3$ in the culture of *S. elongatus*. Nutrients (nitrate, phosphate, silicate, trace metals, and vitamins) were added on the morning of the phytoplankton addition to the tank. The nutrients were mixed into the seawater using the stir bars in the tank (Fig. 1) for approximately an hour before the phytoplankton were added. The maximum potential biomass in the tank was determined by nutrient availability. A low N:P ratio compared with the Redfield Ratio of 16:1 (Redfield, 1958) meant that the cultures were likely to become N-limited rather than P-limited. Silicate was added to both

cultures for consistency, although it is not required by cyanobacteria. In both experiments, trace elements and vitamins were added as in Berges et al. (2001).

**2.1.1 Sample collection and aerosol generation**

Sample collection from the water occurred via a 2 cm port cut into the acrylic lid on the top of the tank. The port was sealed with a silicone rubber bung between water sampling events, which occurred in the morning (1.5 - 2 h after the lights were turned on to simulate daylight). Water was removed from the tank with a 60 mL syringe that was acid-washed in 10% HCl daily and rinsed three times with ultra-high purity (UHP) prior to sampling.

Stirring of the water was switched off during aerosol generation. Immediately prior to aerosol generation, the MART was flushed with HEPA filtered air at 5 Lpm until the number concentration of particles was less than $5 \times 10^{-3}$ L$^{-1}$. The air was turned off, and a peristaltic pump (Watson-Marlow 550Dz) was switched on, providing water flow to a diffuser located $3.56 \times 10^{-1}$ m above the surface of the water. The diffuser was slot shaped, creating an approximately laminar waterfall that impacted the surface of the water. Upon impaction, the waterfall entrains air, generating bubbles which rise to the surface and burst, generating aerosol. The peristaltic pump sent 70 L of water through the waterfall at the top of the tank, which took 30 minutes. Aerosol measurements were taken after a 5 minute equilibration period. Aerosol sampling occurred from a PVC pipe attached to 1.5 cm diameter tubing. The air was dried and passed through a 46 cm long glass mixing tube, with a 0.64 cm I.D. entrance ports on one end, and 0.64 cm I.D exit points on the opposite end. Total aerosol concentration (in the approximate range of ~ 0.01 to ~1 μm diameter) was measured with a water-based Condensation Particle Counter (CPC, TSI, Inc. Model 3786). In addition, a Portable Aerosol Spectrometer (GRIMM 1.108) was used to measure the size-resolved number concentration of 0.3 to 20 μm diameter aerosol. At the beginning of each sampling period, a Gilbrator flowmeter was used to confirm a positive flow outward through the HEPA exhaust filter.

For offline ice nucleation measurements, size-sorted aerosol samples were collected on combusted aluminium foil substrates inside a PIXE cascade impactor with the following stages: 6, 3, and 01, corresponding to 8, 1 and 0.06 μm diameter, respectively (Fig. 1). Samples collected on the L1 impaction stage (0.06-1 μm aerodynamic diameter) were analysed. Aerosol were collected for 2 h at an air flow of 1 L min$^{-1}$ through the sampler. Samples were stored at -80 °C. There were two PIXE cascade impactors in the system (Fig. 1), but only samples from PIXE A were used in the analysis. PIXE B served as a backup system to ensure that back-up samples were available if there were issues such as instrument failure.

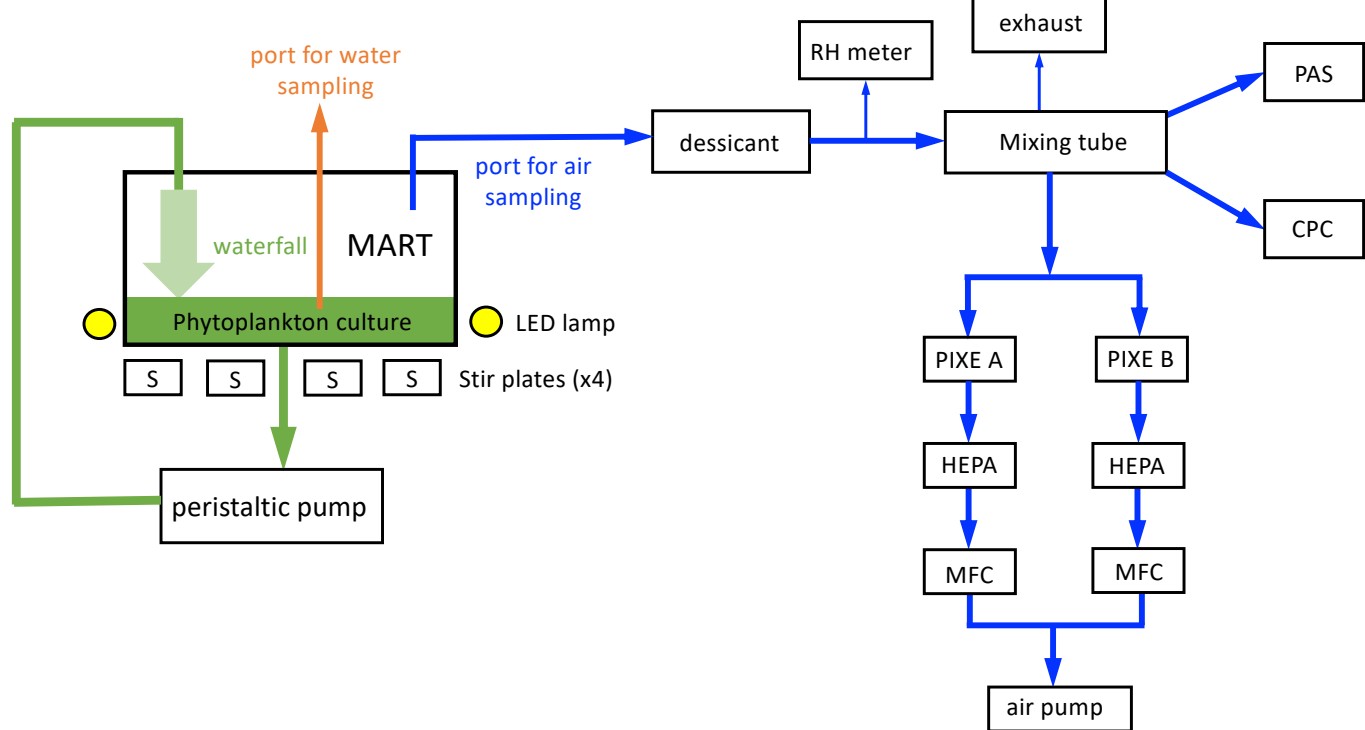

**Figure 1. Marine aerosol reference tank (MART) and associated aerosol sampling. The MART contained a phytoplankton culture (green) and overlying headspace (white). The MART had sampling ports for water sampling (orange) and sampling the headspace above the culture (blue). A peristaltic pump was used to generate a laminar waterfall back into the culture, generating sea spray aerosol (SSA). Light to support phytoplankton growth was provided by LED lamps (170 µmol m$^{-2}$ s$^{-1}$ on a 14 h light: 10 h dark cycle) and the culture was stirred using 4x 6 cm long stir bars.  Air from the headspace of the MART was dried and relative humidity was monitored (RH**
**meter) before being mixed (46 cm long mixing tube). Total aerosol concentration was measured with a water-based Condensation Particle Counter (CPC). A Portable Aerosol Spectrometer (PAS) was used to measure the size-resolved number concentration of aerosol. Size-sorted aerosol samples were collected on combusted aluminium foil substrates inside a PIXE cascade impactor. Filters (HEPA) and mass-flow controllers (MFC) were positioned before the air pump. Arrows indicate the direction of flow.**

## 2.2 Measurement of ice nucleation

The ice nucleation measurements in this study were conducted through off-line ice nucleation experiments using our custom-built ice microscope apparatus (Collier and Brooks, 2016; Fornea et al., 2009; Wilbourn et al. 2020) which allows us to independently determine the freezing temperature for the same sample multiple times. The ice microscope consists of an Olympus optical microscope (Model BX51M), a digital camera (Q-Imaging Micropublisher 5.0 RTV), and Linkam cooling stage (LTS 350).

Temperature control was maintained to within ± 0.1 °C in the sealed Linkam stage throughout freeze-thaw cycles from +5 to -40 °C. As described in detail in Fornea et al. (2009) the stage temperature is routinely calibrated by measuring the melting points of *n*-dodecane, *n*-undecane, and *n*-decane (Collier and Brooks, 2016).

An aerosol sample collected on aluminum foil from the L1 stage of the PIXE cascade impactor was placed on a hydrophobic (silanized by immersion in a 1.0% Aquasil solution (Pierce Chemical Company)) glass microscope slide for support.

To simulate immersion mode nucleation, a 2.0 µL drop of HPLC-grade water (VWR) was micro-pipetted onto the sample containing potential INPs, which were visible as a small circular spot on the aluminum foil. The Linkam stage was sealed and cooled at a rate of 1 °C min$^{-1}$ from +5 °C to -40 °C. Maintaining a dew point of approximately -39 °C prevents droplet evaporation while ensuring that condensation does not form inside the Linkam stage. To do so, a humidified gas flow was generated by combining a flow of water-saturated nitrogen from a glass bubbler containing ultra-high purity (UHP) water (0.01 Lpm) and a flow

of dry nitrogen (0.6 Lpm). Dew point was measured with a hygrometer (EdgeTech DewPrime II, Model 2000). Photographs of the

droplet were taken at 5x magnification every 6 seconds (corresponding to a 0.1 °C temperature change). After the stage reached -40 °C it was heated at a rate of 5 °C min$^{-1}$ until it reached 5 °C, where the temperature was held for 1 minute to ensure complete melting of the droplet. Mean ice nucleation temperature and fraction frozen over the temperature range (0 to -40 °C) were determined from multiple ice nucleation events observed on the same sample by repeating the cooling and warming cycle 25 times. In practice, some runs resulted in a data set with fewer than 25 freezing points due to droplet evaporation. Images were analyzed on a frame-by-frame basis to visibly determine the freezing temperature. Data from the measurement of a single sample in the Linkam stage were only included in further analysis if the following criteria were met: 1) a discrete freezing event was observed in a minimum of three freeze-thaw cycles, 2) freezing was clearly observed at a discrete temperature for each observation (i.e. in a single CCD image), 3) the droplet did not visibly change size, and 4) no condensation was observed adjacent to the droplet on the slide. In procedural blank runs, homogeneous droplets of HPLC-grade water (VWR Scientific) were observed at -33.5 ± 2.0 ºC on combusted aluminium foil (used as substrates in the PIXE sampler) and at -33.1 ± 0.6 ºC on silanized glass slides).

The cumulative number concentration of INPs per L of air in the headspace of the MART ($N_{INP}$) was calculated using the following equation (DeMott et al., 2016; Vali et al., 1971):

$$N_{INP}(T) = -\ln\frac{N_u(T)}{N_0} * \frac{V_w}{V_a \times V_s} \qquad (1)$$

where $N_u(T)$ is the cumulative number of unfrozen droplets at a given temperature and $N_0$ is the total number of freezing events for a sample compiled from all replicates. $V_a$ is the volume of a single aliquot (2 μL for this study), $V_w$ is the total volume of all water droplets (calculated by multiplying $V_a$ by the number of total freezing events from all replicates), and $V_s$ is the total volume of air sampled on the impactor.

## 2.3 Phytoplankton biomass, growth and physiology

### 2.3.1 Phytoplankton biomass

Chlorophyll *a* concentrations were determined from water samples filtered onto GF/F filters (Whatman, GE Healthcare Bio-Sciences) and stored at -20°C until analysis. Chlorophyll *a* was determined using a spectrophotometric method (Parsons et al., 1984) for the experiment with *T. weissflogii* (Jeffrey and Humphrey, 1975), and a fluorometric method was used in the experiment with *S. elongatus* (Arar and Collins, 1997). For both protocols, pigments were extracted with 90% acetone and sonication was used to break up the cells. Samples were sonicated (Qsonica, Q125 [125 Watts, 20 kHz]) for 10 min with the amplitude set at 40% in 5 s pulses. Heat build-up was prevented by keeping the extractions on ice with 5 s pauses between pulses. Pigment extraction was continued in the dark overnight at 4°C and the extractions were centrifuged at 1,000 *g* for 20 min at 4°C prior to measurement of chlorophyll *a* concentration in the supernatant.

Samples of water (1 mL) were collected from the MART tank in triplicate and placed in small glass vials for cell counts from the culture of *T. weissflogii*. The samples were fixed with a drop of acidic Lugol's iodine (Parsons et al. 1984) and counted by light microscopy using a hemacytometer (Fuchs-Rosenthal ruling, Hauser Scientific) to count 400 cells (Guillard and Sieracki, 2005). Total (including both cellular and extracellular) carbohydrate concentrations in the medium were measured daily during the experiment with *T. weissflogii*. Carbohydrates were measured using the phenol-sulfuric acid assay (Dubois et al., 1956) calibrated with D-glucose. Culture (0.8 mL) was placed in a boiling tube mixed with 0.5 mL of 5% (w/v) phenol. Concentrated sulfuric acid (2 mL) was added rapidly using a dispenser bottle and absorbance was measured at 485 nm after 20 minutes.

### 2.3.2 Physiological status

Variable chlorophyll fluorescence was used as an indicator of physiological status of the phytoplankton in the MART. The maximum quantum yield of photosystem II ($\phi_{PSII}$) was measured using the saturating pulse method (Genty et al., 1989; Maxwell and Johnson, 2000). This parameter measures the proportion of light absorbed by the photosynthetic pigment chlorophyll in photosystem II that is used to drive photochemistry (Maxwell and Johnson 2000). Measurements were made on dark adapted (30 minutes) samples using pulseamplitude modulated (PAM) fluorescence (PHYTO-PAM fitted with a PHYTO-ED emitter-detector unit (Heinz Walz GmbH)). The four channels of the PHYTO-PAM have different excitation wavelengths (470, 520, 645 and 665 nm) and the detector is protected by a long-pass filter, measuring emission at wavelengths > 710 nm. Data produced by excitation at 665 nm was used in this study. The instrument was set up with a measuring light pulse frequency of 2 (equivalent to photosynthetically active radiation (PAR) of 1 $\mu$mol m$^{-2}$ s$^{-1}$), a saturation pulse intensity setting of 10 and a pulse width of 200 ms. $\phi_{PSII}$ was calculated based on variable fluorescence ($F_v$), where $F_o$ is the minimum fluorescence yield of the dark-adapted sample when all reaction centers II are open, and $F_m$ is the maximum fluorescence yield when all reaction centers II are closed (Genty et al., 1989; Maxwell and Johnson, 2000):

$$\phi_{PSII} = F_v/F_m = (F_m - F_o)/F_m \qquad (2)$$

Cell membrane permeability and therefore potential cell 'leakiness' was determined from counts of *T. weissflogii* stained with SYTOX Green (Invitrogen, Thermo Fisher Scientific). SYTOX Green is a plasma membrane impermeable nucleic acid dye (Veldhuis et al., 2001; Franklin et al., 2012) that stains cells with a compromised plasma membrane, causing fluorescence with an emission peak of 523 nm when excited by a source at 450–490 nm. Samples (1 mL) were placed in sterile microcentrifuge tubes and incubated with 1.92 $\mu$M SYTOX Green stain for 1 hin the dark at 20 °C. Stained sample (0.5 mL) was mixed with 1.5 mL of filter sterilized ASW and filtered through a 0.4 $\mu$m pore size polycarbonate filter (Nuclepore, Whatman) under low vacuum (< 150 mm Hg). Excess dye was removed by two rinses (1 mL) with filter sterilized ASW. The filters were mounted on glass slides using SlowFade Diamond Antifade Mountant (Invitrogen, Thermo Fisher Scientific). Cells (400 per sample) were counted using an epifluorescence microscope (Axioplan 2, Carl Zeiss MicroImaging) and categorized depending on how they stained (Thornton, 2014): intact plasma membrane (unstained), compromised plasma membrane (stained nucleus), and compromised plasma membrane with breakdown of internal membranes (stained throughout the cells).

### 2.3.3 Extracellular carbon pools

Two classes of exopolymer particles were measured in the MART. Transparent exopolymer particles (TEP) and Coomassie stainable particles (CSP) are ubiquitous in the ocean (Thornton, 2018) and are often associated with phytoplankton growth (Passow, 2002; Engel et al., 2015; Thornton, 2018) and death (Berman-Frank et al., 2007; Bouchard et al., 2011; Thornton and Chen, 2017). Exopolymer particles were collected by filtration (2 mL) under low vacuum onto 25 mm diameter 0.4 $\mu$m pore size polycarbonate filters (Nuclepore, Whatman, GE Healthcare Bio-Sciences) in triplicate. The filters were stained directly in the filter funnel with 1 mL of dye that was drawn through the filter slowly, followed by two rinses of 1 mL of UHP water to remove excess dye. TEP were stained with Alcian blue 8GX (Sigma-Aldrich), which is specific for acid polysaccharides. The Alcian blue solution was 0.02% (w/v) in 0.06% acetic acid (v/v) at pH 2.5 (Passow and Alldredge, 1995). CSP were stained on separate filters with

Coomassie Brilliant Blue G-250 (Sigma-Aldrich), which is specific for proteins (Long and Azam, 1996). Working Coomassie Brilliant Blue G-250 solutions were prepared each day by making a 1/25 dilution of the stock solution (1 g Coomassie Brilliant Blue G-250 in 100 mL of UHP water) with 0.2 μm filtered ASW (Berges et al., 2001). This resulted in a 0.04% (w/v) working solution at pH of 7.4. Dry filters were mounted in a drop of immersion oil on a Cytoclear microscope slide (GE Water & Process Technologies; Logan et al., 1994). A second drop of oil was placed on top of the filter, and it was covered with a glass coverslip. Prepared slides were stored frozen (-20°C).

Exopolymer particles were quantified in terms of particle abundance (particles mL$^{-1}$) and concentration (mm$^2$ mL$^{-1}$) from JPEG images of the filters, captured using an AxioCam ERc 5s color camera mounted on an Axioplan 2 microscope run from Axiovision 4.8 software (Carl Zeiss MicroImaging). The image analysis method was based on Engel (2009), see Thornton and Chen (2017) for details. Briefly, image analysis was conducted using ImageJ software (National Institutes of Health). Only the red channel of the JPEG image was analysed, using the triangle method (Zack et al., 1977) to threshold the grayscale image into a binary image where TEP or CSP were represented by black pixels.

Water samples were filtered through combusted glass fiber filters (GF/F, Whatman) and stored in combusted glass vials at -20°C in the dark for analysis of fluorescent dissolved organic matter (FDOM). Samples were thawed slowly at room temperature in the dark and placed in a 1 cm path quartz cuvette. Absorbance spectra and excitation-emission matrices were measured using a Horiba Aqualog fluorometer with excitation wavelengths from 240 to 500 nm in 2 nm increments. A high-purity water sample (Starna Scientific Ltd.) was used to correct fluorescence and absorption spectra. Fluorescence spectra were corrected by inner filter effect using the Aqualog software, and intensities were converted to Raman units (excitation at 350 nm, emission intensity from 371 to 428 nm) (Kothawala et al., 2013; Lawaetz and Stedmon, 2009). FDOM was characterized using Coble's peaks (Coble, 1996, 2007). These peaks in fluorescence spectra are associated with defined chemical properties of DOM, though they cannot be assigned specific chemical formulae (Coble, 1996). We measured peak B (tyrosine-like, protein like; emission intensity at 310 nm, with an excitation of 275 nm), peak T (tryptophan-like, protein-like; emission intensity at 340 nm, with an excitation at 275), peak A (humic-like; peak emission intensity between at 380 to 460 nm, with an excitation at 260 nm); peak M (humic-like; peak emission intensity between at 380 to 420 nm, with an excitation at 312 nm), peak C (humic-like; peak emission intensity between at 420 to 480 nm; with an excitation at 350 nm).

### 2.3.4 Bacterial activity

Glucosidase activity was used as a proxy for bacterial metabolism during the MART experiment with *T. weissflogii*. Enzyme activity was measured using the fluorimetric method of Hoppe (1983) modified for a 96-well plate format (Marx et al., 2001). The substrate analog for β-glucosidase activity was 4-methylumbelliferone-β-D-glucoside (Sigma-Aldrich), which is cleaved by β-glucosidase to produce a fluorescent product, 4-methylumbelliferone (MUF). Each assay (200 μL) consisted of 180 μL of water from the MART and 20 μL of substrate. The assay was intentionally not conducted in a biological buffer to maintain the pH observed within the MART. Temperature within the plate reader varied < 1.5 °C between assays on different days. Substrate concentration was 200 μM, which preliminary work showed to be saturating; therefore the enzyme velocities measured represent β-glucosidase activity. Fluorescence intensity was measured using a Spectramax Gemini EM spectrofluorometer run by SoftMax Pro 4.8 software. The plate (opaque white 96 well (Grenier)) was mixed by vibration for 5 seconds prior to each measurement. Fluorescence intensity was measured every 2.5 minutes for up to 12 hs, however only the first 120 minutes of data were used to calculate rates as this was the period in which there was a linear increase in fluorescence, indicating that the substrate was not limiting. The assay was calibrated using a dilution series of MUF standards, with calibrations measured several times over the

course of the MART experiment. Heat killed controls (MART water heat to 90-95 °C for 15 minutes) were run as well as ASW and UHP water blanks. Samples from the MART were size fractionated by filtration to locate the enzyme activity; in addition to whole water, the assay was run on water passed through GF/C filter (Whatman) with a nominal pore size of 1.2 μm, and double-filtered through a GF/C filter followed by a 0.2 μm pore size syringe filter. Whole water samples contained phytoplankton, bacteria and detritus. Passage through a GF/C filter was used to remove *T. weissflogii*, detritus and particle associated bacteria. Passage through a 0.2 μm pore size filter was used to remove free-living bacteria in addition to *T. weissflogii* and detritus.

**2.4 Artificial Seawater blank MART**

A MART experiment was conducted to determine the background conditions in a tank containing artificial seawater, but no phytoplankton. This experiment represented a controlled blank, which was used to make comparisons to the experiments during which phytoplankton were grown. The experiment was conducted over 5 days, using the same protocols as the experiments inoculated with phytoplankton. The experiment was conducted in two phases; firstly, the tank was filled with ASW (no nutrient or phytoplankton additions) and the aerosol were generated and sampled as described above on days 1 and 2. Secondly, L1 nutrients were added on day 3 and the measurements were repeated on days 3 and 5. The blank measurements were broken down into these two phases due to the compositional difference between ASW and ASW+L1, as L1 nutrients contain sources of macronutrients (N, P and Si), trace metals, and also sources of organics (Guillard and Hargraves, 1993) in the form of $Na_2EDTA$ (14.9 μM) and vitamins (thiamine (0.3 μM), biotin (2.05 x $10^{-3}$ μM), and cyanocobalamin (3.69 x $10^{-4}$ μM). The concentrations of macronutrients added were 60 μM $NaNO_3$, 20 μM $NaH_2PO_4$, and 60 μM $Na_2SiO_3$. Aerosol concentration and ice nucleation properties were measured. Biological measurements were chlorophyll *a* concentration, bacteria counts, and TEP and CSP concentrations. Ice nucleation measurements made on samples collected during this experiment were considered procedural blanks.

**2.5 Field methods**

Results from the MART experiments were compared with ice nucleation data collected *in situ* from the *R/V Atlantis* in the North Atlantic during the second NASA North Atlantic Aerosols and Marine Ecosystems Study (NAAMES) research cruise in May 2016 (Wilbourn et al., 2020). The objective of NAAMES was to understand the ocean ecosystem-aerosol-cloud system of the western subarctic Atlantic, including how plankton ecosystems influence the properties of remote marine aerosol and boundary layer clouds (Behrenfeld et al., 2019, 2021). The data were collected at a single station (Station 4; 47◦ 39.360 N, 39◦ 11.398 W), after a storm caused deep water entrainment with homogenous physical properties to well below 200 m on arrival at the station, followed by shoaling resulting in a mixed layer < 25 m within 24 to 48 hs (Graff and Behrenfeld, 2018). Drifters and float data were used to maintain the sampling station within the same water mass over the next 4 days (Graff and Behrenfeld, 2018). Della Penna and Gaube (2019) defined Station 4 as a subtropical (based on mean dynamic topography) anticyclonic eddy.

Detailed methods are provided in Wilbourn et al., (2020). Briefly, high performance liquid chromatography (HPLC) was used to quantify phytoplankton pigments (Van Heukelem and Thomas, 2001) at the NASA Goddard Space Flight Center Ocean Ecology Laboratory (Greenbelt, Maryland). Pigments were filtered onto combusted GF/F filters, stored at -80 °C, extracted in methanol, and analyzed on a 4.6 x 150 mm HPLC Eclipse XDB column using an Agilent RR1200 HPLC system (Agilent Technologies). In addition, small phytoplankton were enumerated and grouped using a cell sorting flow cytometer (BD Influx, Becton Dickinson Biosciences) (Graff et al., 2012, 2015). The size ranges 0.5-1 μm, 1-3 μm, 3-50 μm corresponded to three groups of microorganisms: *Synechococcus*, photosynthetic picoeukaryotes, and photosynthetic nanoeukaryotes. Cells were sorted based on pigment fluorescence emissions at 692 nm and 530 nm, and forward and side scattering intensity.

Primary marine aerosol were generated from the surface of the ocean using the Sea Sweep aerosol generator (Bates et al., 2012, 2020). Aerosol sample collection using PIXE impactors, and ice nucleation measurements were conducted employing the technique described above for the mesocosm study. Air sample flow (1 Lpm) from the Sea Sweep was directed to the PIXE cascade impactor to collect 0.06-1 µm aerodynamic diameter on combusted aluminium foil. Aerosol samples were stored at -80 °C and transported to Texas A&M for ice nucleation measurements (Wilbourn et al., 2020) as described above.

## 3 Results

### 3.1 Phytoplankton culture experiments in a marine aerosol reference tank (MART)

Data collected during the 24 day MART experiment with *S. elongatus* are summarized in Fig. 2. Growth rates of *S. elongatus* per day and mean INP freezing temperatures are shown in Fig. 2a. In this experiment, growth rate was determined from changes in

chlorophyll *a* concentration, a metric commonly used to track changes in phytoplankton biomass (Fig. 2b). Growth rate changed from days 2 to 4 was 1.34 day$^{-1}$, associated with a peak biomass of 34.1 ± 0.2 µg chl. *a* L$^{-1}$ (mean ± SD; $n = 3$). The chlorophyll *a* concentration declined between measurements made on days 4 and 8, resulting in negative growth rates during this period, indicative of a dying culture. There was a second period of growth during which the biomass of *S. elongatus* increased from mean values of 15.3 to 26.0 µg chl. *a* L$^{-1}$, but the growth rate was relatively low (0.21 day$^{-1}$) compared with the initial period of relatively

fast growth (1.34 day$^{-1}$). Mean ice nucleation freezing temperatures of aerosol collected from the MART were warmest during the initial period of high growth rates, with freezing temperatures of -23.9 ± 1.9 and -18.0 ± 2.3 °C (mean ± SD) on days 2 and 3, respectively. These temperatures were significantly warmer than homogeneous freezing of pure water droplets, indicating the presence of effective INPs in the aerosol. Mean ice nucleation freezing temperatures for the remainder of the experiment were relatively cold, ranging from -34.2 ± 1.1 °C (day 4) to -30.7 ± 1.1 °C (day 8). Ice nucleation freezing temperature measurements of

aerosol collected in the blank MART experiment represent procedural blanks, as the samples were treated in the same way as samples from the MART experiments with phytoplankton. The mean ice nucleation freezing temperature of aerosol collected from the headspace of a blank MART containing only ASW was -34.9 ± 0.9 °C, compared with -31.3 ± 0.9 °C for ASW+L1 nutrients (Fig. 2a). As described in the methods, the addition of L1 nutrients included organics, mainly in the form of vitamins.

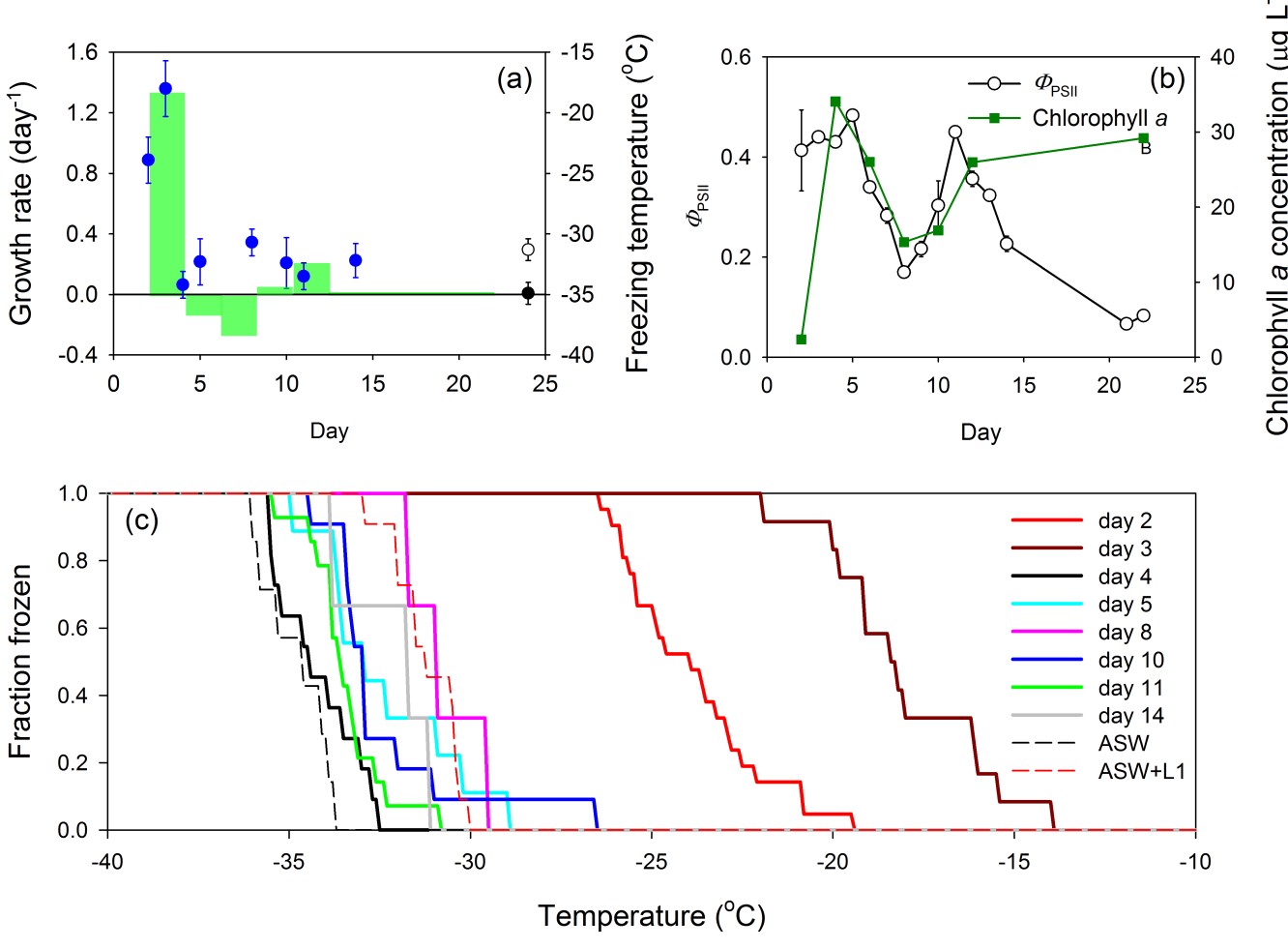

Figure 2. Growth of the cyanobacterium *S. elongatus* in a marine aerosol reference tank (MART) and associated ice nucleation. (a) Growth rates of *S. elongatus* (per day) (green bars) with time (indicated by the width of the green bars), plotted with mean freezing temperature of ice nucleating particles (INPs) (blue circles) over time (data points indicate mean ± pooled SD). The freezing temperature of aerosol procedural blanks are shown for reference, where the black circle represents the freezing temperature of aerosol generated from artificial seawater medium (ASW) (mean ± SD, n = 7) and the open circle represents the freezing temperature of aerosol generated from ASW + L1 nutrients (mean ± SD, n = 11). Procedural blank measurements were made during a separate blank MART experiment and therefore their position on the x-axis does not indicate timing. (b). Quantum yield of photosystem II ($\Phi_{PSII}$) was used to indicate the physiological status (mean ± SD, n = 3) of the cyanobacterium (open circles) over time, while changes in chlorophyll *a* show changes in biomass (green squares) (mean ± SD, n = 3). (c) The fraction of INPs frozen at different temperatures in aerosol collected from the MART. Each solid line represents data from a different day. Procedural blanks are shown as dotted lines for artificial seawater (ASW) and artificial sea water with added nutrients (ASW+L1).

The quantum yield of photosystem II ($\phi_{PSII}$) is a measure of how efficiently phytoplankton use light energy during photosynthetic photochemistry and it is frequently used as an indicator of general physiological status (Fig. 2b). The $\phi_{PSII}$ was relatively high during the first few days of growth in the MART, increasing from 0.41± 0.08 (day 2) to 0.48 ± 0.0 (day 5) (mean ± SD; n = 3). The standard deviations of measurements of both $\phi_{PSII}$ and chlorophyll *a* were smaller than the data points in most measurements in Fig. 2b. Correlating with changes in chlorophyll concentrations, $\phi_{PSII}$ declined between days 6 and 8, and increased during the second period of growth from 0.17 ± 0.01 (day 8) to 0.45 ± 0.01 (day 11) (Fig. 2b). On days 21 and 22, $\phi_{PSII}$

was very low (< 0.1) indicating that the *S. elongatus* culture was dying, though chl. *a* concentration remained high (29.2 ± 0.1 µg chl. *a* L$^{-1}$) (Fig. 2b). The decline in biomass and $\phi_{PSII}$ (days 5 to 8), followed by a second growth period and increasing $\phi_{PSII}$ between days 8 and 11, was unexpected. There was no change in environmental conditions within the MART to explain the observed pattern.

    The fraction of INPs frozen as a function of temperature is shown in Fig. 2c. The onset of freezing and the temperature of complete

INP activation were both much warmer early in the experiment (days 2 and 3) than for aerosol collected on subsequent days. On day 2, INPs in the aerosol started to activate at -19.5 ºC and were completely activated at -26.5 ºC (Fig. 2c). Activation started at the warmer temperature of -14.0 ºC on day 3 and all INPs were activated at -22.0 ºC. In contrast, activation occurred at colder temperatures, between -26.6 and -33.5 ºC, for the later sampling days (Fig. 2c). There was no relationship between the biomass of *S. elongatus* in the tank and the concentration of aerosol in the overlying air (Fig. S1). Plots of fraction frozen against freezing

temperature show that, except for warm freezing temperatures on days 2 and 3, the freezing temperatures of aerosol from the *S. elongatus* culture were between those of the procedural blanks; generally warmer than ASW on its own, but colder than ASW+L1. Therefore, except for days 2 and 3, there was no difference between the freezing temperature of SSA containing phytoplankton organic matter and the procedural blanks.

The diatom *T. weissflogii* showed the same pattern as *S. elongatus*, with fast growth rates associated with ice nucleation at the warmest temperatures (Fig. 3a), with a rapid increase chlorophyll *a* concentration (38.7 to 188.9 µg L$^{-1}$ between days 1 and 3) and relatively high values of $\phi_{PSII}$ (> 0.5) (Fig. 3b). The standard deviations of measurements of both $\phi_{PSII}$ and chlorophyll *a* were smaller than the data points in most measurements in Fig. 3b. The onset of ice nucleation on days 2 and 3 were -19.1 ºC and -22.3 ºC, respectively (Fig. 3c). On these days, the fraction of INPs frozen reached 100% at -24 ºC (Fig. 3c). In contrast, the onset

of nucleation was < -27 ºC for sampling days at slower growth rates later in the experiment (Fig. 3c). Fraction frozen curves showed that freezing occurred at temperatures significantly warmer than the procedural blanks at the beginning of the experiment (days 2 and 3) and at the end of the experiment (days 16 and 29). Several improvements to the experimental procedure were implemented for the *T. weissflogii* experiments. A higher temporal resolution of growth rate estimates was determined on a day-to-day basis using direct cell counts (Fig. 3d). The highest growth rate in the cultures was 3.0 day$^{-1}$ observed on day 1, but no

samples for INP characterization were taken on that day. On days 2 and 3, growth rates were 1.0 and 0.5 day$^{-1}$, respectively, corresponding freezing temperatures of -20.7 ± 1.4 ºC and -23.0 ± 0.7 ºC (Fig. 3a). Total carbohydrate increased rapidly between days 2 (13.9 ± 1.1 µg mL$^{-1}$; mean ± SD, $n = 3$) and 5 (20.5 ± 0.5 µg mL$^{-1}$), corroborating the pattern in biomass and growth rates observed from cell counts (Fig. 3d). Day 3 was the day on which the chl. *a* concentration was highest (188.9 ± 4.8 µg chl. *a* L$^{-1}$; mean ± SD, $n = 3$) (Fig. 3b), with a cell count of 7.58 x 10$^4$ ± 4.39 x 10$^3$ cells mL$^{-1}$ (mean ± SD, $n = 3$) (Fig. 3d). Growth rates on

400     days 7 and 9 were 0 day$^{-1}$, indicating stationary phase. Based on cell counts, there was a significant correlation between growth rate over the 24 hs before the sample was taken and mean nucleation temperature (r = 0.93, p < 0.001, n = 20). There was a correlation (r = 0.917, p < 0.001, n = 13) between $\phi_{PSII}$ and growth rate in *T. weissflogii*, with high $\phi_{PSII}$ indicating photosynthetically efficient cells that were growing rapidly. High values of $\phi_{PSII}$ (> 0.44) were associated with ice nucleation at warmer temperatures (> -24 ºC) during the *T. weissflogii* experiment. SYTOX Green staining was measured as a further indicator of the physiological

status of the *T. weissflogii* (Fig. 3e). It is a marker of cell permeability, in which the increase in staining indicates that cells are potentially leaking more DOM into the surrounding water. In the *T. weissflogii* experiment the proportion of cells that stained with the fluorescent probe increased rapidly as the concentration of diatoms decreased in the MART, from < 5 % during the first seven days of culture, to 11 ± 5 % (mean ± SD) on day 9, and 92 ± 2 % by day 13.

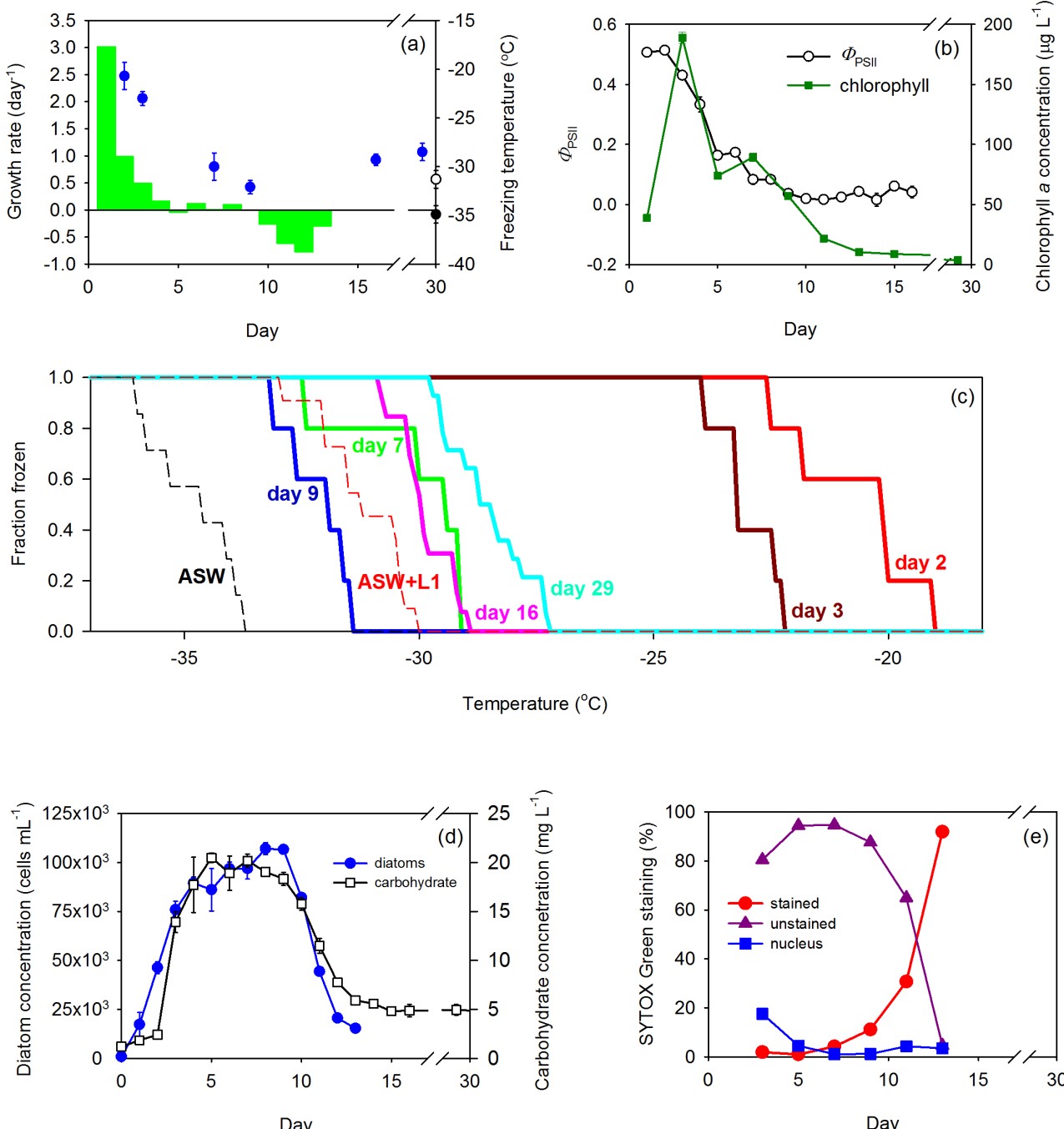

**Figure 3. Growth of the diatom *T. weissflogii* in a marine aerosol reference tank (MART) and associated ice nucleation. (a).** Day-to-day growth rates (per day) of *T. weissflogii* determined based on changes in cell concentration (green bars) plotted with mean freezing temperature of ice nucleating particles (INPs) (blue circles) over time (data points indicate mean ± pooled SD). Growth rates were measured for the first 13 days of the experiment. The mean freezing temperature of aerosol procedural blanks are shown for reference, where the black circle represents the freezing temperature of aerosol generated from artificial sweater medium (ASW) (mean ± SD, n = 7) and the open circle represents the freezing temperature of aerosol generated from ASW + L1 nutrients (mean ± SD, n = 11). Blank measurements were made during a separate blank MART experiment and therefore their position on the x-axis does not indicate timing. **(b).** Quantum yield of photosystem II ($\Phi_{PSII}$) was used to indicate the physiological status (mean ± SD, n = 3) of the diatom (open circles) over time, while changes in chlorophyll *a* show changes in biomass (green squares) (mean ± SD, n = 3). **(c).** Fraction of INPs frozen and concentration of INPs in the atmosphere above the culture in the MART at different temperatures. The coloured solid lines represent different days. Procedural blanks are shown as dashed lines for artificial seawater (ASW) and artificial sea water with added nutrients (ASW+L1). **(d).** Changes in diatom biomass within the culture over time based on diatom cell abundance (mean ± SD, n = 3) (blue circles) and total carbohydrate (open squares) (mean ± SD, n = 3). **(e)** SYTOX Green staining as an indicator of cell leakiness and death over

time. The proportion of diatoms that were unstained and therefore had intact cells (purple triangles), partially stained with a stained nucleus (blue squares), or fully stained indicating dying or dead cells (red circles).

Concentrations of INPs in the headspace above both cultures are shown in Fig. 4. Concentration of INPs in both experiments were similar, despite cultures of *T. weissflogii* reaching a biomass (Fig. 3b) an order of magnitude greater than that of *S. elongatus* (Fig. 3a), as indicated by chlorophyll *a* concentration. The maximum concentration of INPs in both experiments was approximately $2 \times 10^{-3}$ INP L$^{-1}$ air at -26 ºC (*T. weissflogii*) and -29 ºC (*S. elongatus*). INPs were observed at the warmest temperatures on day 3 in cultures of *S. elongatus*, with the onset of nucleation first detected ($6 \times 10^{-5}$ INPs L$^{-1}$) at -14 ºC .

Information on additional variables which were assessed, but did not indicate a clear relationship to ice nucleation behavior, is provided in the Supplement. Aerosol number concentrations in the MART were greater in the headspace over cultures (Fig. S1) compared with the headspace over blank seawater (Table S1), indicating that biomass and biological activity affected the production of aerosol. Aerosol number concentration was generally higher (mean = $2.21 \times 10^6$ L$^{-1}$) over the culture of *T. weissflogii* compared with *S. elongatus* (mean = $1.43 \times 10^6$ L$^{-1}$), reflecting the relatively higher biomass in the diatom culture, as indicated by chlorophyll *a* concentration in Figs. 2b and 3b above. Less aerosol were generated from a control MART tank containing only artificial seawater ($< 1 \times 10^6$ L$^{-1}$; Table S1). Aerosol number concentration increased in the control experiment after the addition of L1 nutrients, possibly to the addition of organic matter in the form of vitamins, and the growth of bacteria in the water (Table S1). In summary, there is no connection between aerosol concentration and ice nucleation activity indicating that the improvements in ice nucleating ability are not driven by the total aerosol available to catalyze freezing.

Results from the analysis of the extracellular material in the MART experiments, namely exopolymer particles (TEP and CSP) and FDOM is provided in the supplementary material (Figures S2,S3, and S4). DOM in the medium provides potential substrates for bacterial metabolism and growth. Measurements of ß-glycosidase activity were used as a proxy for bacterial activity in the *T. weissflogii* experiment (Figure S5). ß-glycosidase activity increased by two orders of magnitude over the course of the experiment, indicating that bacteria were playing a role in the transformation and remineralization of organic matter in the MART. Filtration removed glucosidase activity, indicating that the enzyme was associated with cells or aggregates of organic matter and there was no significant extracellular glucosidase in the culture medium. Diatom biomass (Fig. 3d) declined from day 8 onwards; the continued increase in glucosidase activity suggests that the enzyme activity was associated with heterotrophic bacteria and the remineralization of organic matter as the phytoplankton died.

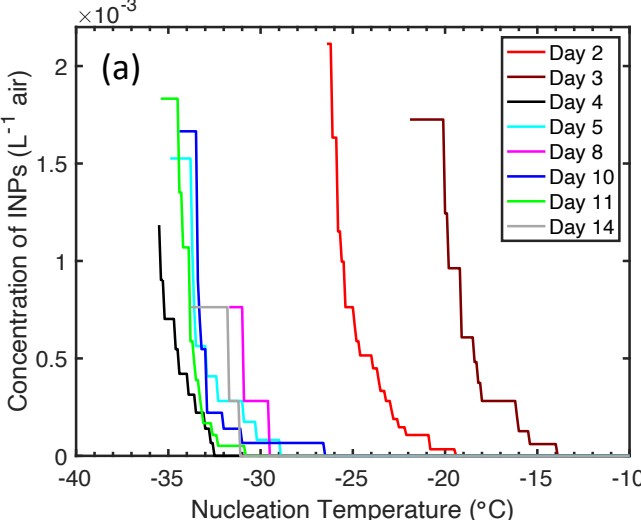
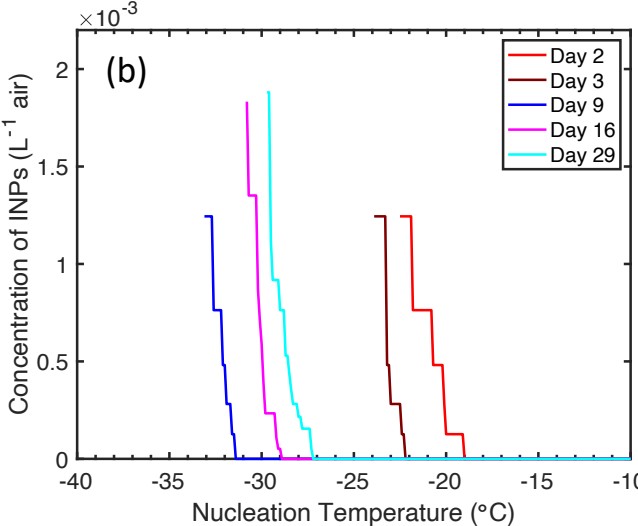

**Figure 4. Concentration of ice nucleating particles (INPs) in the headspace of a marine aerosol reference tank (MART) at different temperatures. Measurements were made over time on different days in batch cultures of (a) *S. elongatus* (cyanobacterium) and (b) *T. weissflogii* (diatom).**

## 3.2 Field observations from the North Atlantic

Field data from Wilbourn et al., (2020) were used to calculate phytoplankton growth rates to determine whether high growth rates were associated with ice nucleation at relatively warmer temperatures. Fig. 5a shows changes in phytoplankton biomass in the surface water of the North Atlantic Ocean, determined in this case, from pigments and flow cytometry. As Station 4 was a natural ecosystem there was grazing (Morison et al., 2019), which was absent during the MART experiments. Consequently, changes in biomass at Station 4 reflect the loss of biomass through grazing, as well as additions through phytoplankton growth. Net growth occurred at Station 4, with an increase in chlorophyll *a* concentration from 0.52 to 1.28 mg m$^{-3}$ on days 1 and 4, respectively. Cell counts corroborated an accumulation in phytoplankton biomass, with concentrations of *Synechococcus* increasing from $2.34 \times 10^6$ to $1.38 \times 10^7$ cells L$^{-1}$. Changes in biomass between days were used to calculate net growth rates (Table 1) at Station 4. Total phytoplankton growth rate, based on change in chlorophyll *a*, was greatest between days 3 and 4, with a rate of 0.56 day$^{-1}$ (Fig. 5b). Growth rate of *Synechococcus* (Table 1) was greatest between days 3 and 4, though it is not known how much the rapid increase in *Synechococcus* contributed to the total increase in phytoplankton biomass. Measurements showed that mean ice nucleation temperatures decreased from day 1 to 3 on station 4 (Fig. 5b), and were warmest at -24.5 ± 0.9 °C (mean ± SD) on day 4, which was associated with the highest phytoplankton biomass (Fig. 5a) and highest net phytoplankton growth rate over the preceding 24 hs (Fig. 5b). The fraction of INPs as a function of temperature is shown in Fig. 5c. The onset ice nucleation temperature and the temperature of complete INPs activation was much warmer on day 4 (Fig. 5c). Taken together, our MART tank and NAAMES experiments shown that in two very different taxa grown in the laboratory, and in complex natural phytoplankton assemblages, the highest propensity to nucleate atmospheric ice crystals occurs during conditions of the fastest growth rates.

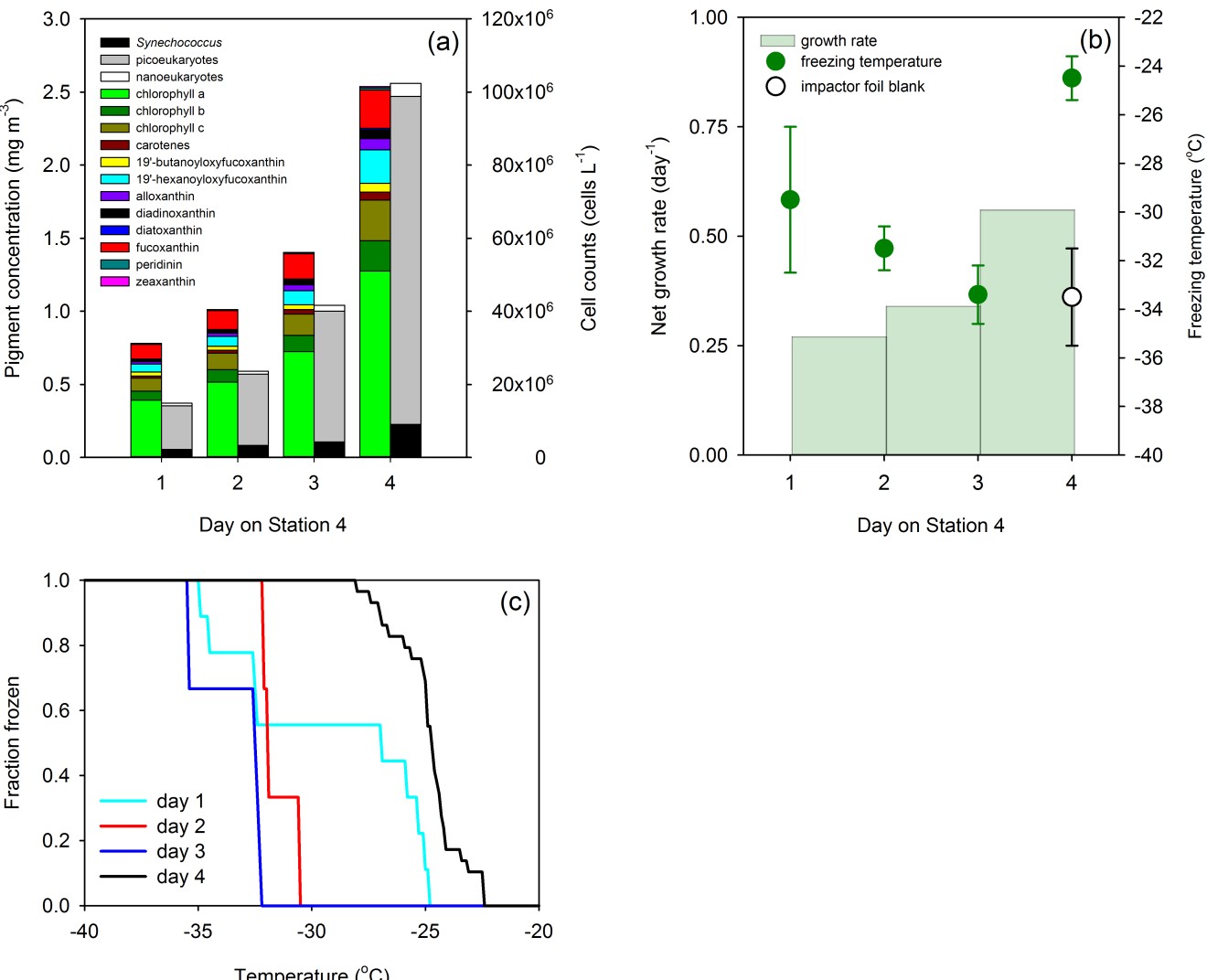

**Figure 5. Phytoplankton growth and ice nucleation over 4 days at a Lagrangian station in the North Atlantic Ocean (24 to 27 May 2016).**
**(a) Changes in phytoplankton biomass as indicated by pigment concentrations (colored stacked bars) and cell abundance (monochrome stacked bars). (b) Day-to-day growth rates determined from changes in chlorophyll *a* concentration and ice nucleation temperature of primary marine aerosol generated *in situ* from surface water (data points indicate mean ± pooled SD) (c) Fraction of droplets frozen at different temperatures in primary marine aerosol generated *in situ*. Each coloured line represents aerosol collected on a different day. Field data replotted from Wilbourn et al. (2020), with the addition of growth rates.**

| Time period (days) | Growth rate (day$^{-1}$) | | | |
| --- | --- | --- | --- | --- |
| | chl. *a* | *Synechococcus* | picoeukaryotes | nanoeukaryotes |
| 1-2 | 0.27 | 0.29 | 0.54 | 1.02 |
| 2-3 | 0.34 | 0.37 | 0.65 | 0.60 |
| 3-4 | 0.56 | 1.12 | 0.68 | 0.57 |

**Table 1. Net phytoplankton growth rates (day$^{-1}$) at a Lagrangian station tracked over 4 days (24 to 27 May 2016) in the North Atlantic Ocean. Growth rates calculated based on changes in biomass concentration over 24 h. Chlorophyll *a* was used as a proxy for total phytoplankton biomass. Different components of the pico- and nanophytoplankton were counted using flow cytometry. Growth rates calculated using data from the North Atlantic Aerosol Marine Ecosystem Study (NAAMES) (Behrenfeld et al. 2019; Wibourn et al. 2020).**

## 4 Discussion

Our results show that SSA generated from cultures of two contrasting taxa (*T. weissflogii* (diatom) and *S. elongatus* (cyanobacterium)) promoted immersion mode ice nucleation at temperatures warmer than the homogeneous freezing of liquid water droplets. Several previous studies have shown that phytoplankton promote ice nucleation (Knopf et al., 2011; Alpert et al., 2011a, 2011b; Wilson et al., 2015; McCluskey et al., 2017, 2018; Wolf et al., 2019; Wilbourn et al., 2020). The unique finding of this study is that the physiological status of phytoplankton in the water affects the properties of SSA sufficiently to increase their ice nucleation temperature. Fast growing phytoplankton produced SSA containing INPs that catalysed ice nucleation at temperatures that were warmer than during other phases of growth, including stationary phase and death phase. This is a finding that was observed in two contrasting taxa and confirmed by a re-examination of field data from Wilbourn et al. (2020). In both the *S. elongatus* and *T. weissflogii* experiments, periods of elevated ice nucleation ability coincided with high $\Phi_{\text{PSII}}$. Further work is needed to determine whether $\Phi_{\text{PSII}}$ could be developed as a reliable indicator of ice nucleation activity. The data presented here suggests that the physiology of fast-growing phytoplankton is associated with the production of a chemical signature (a specific compound, class of compounds, or even a change in the ratio of compounds) that affects the ice nucleation of aerosols at relatively warm temperatures.

It is well established that phytoplankton acclimate to different growth conditions (Geider et al., 1997, 1998; Moore et al., 2006), resulting in different chemical compositions (Geider and LaRoche, 2002; Dyhrman et al., 2012; Lin et al., 2016). While it is impossible to screen all the different biomolecules within a taxon for ice nucleation activity, observations of resource allocation and resource allocation models provide a conceptual framework to identify differences in the composition of phytoplankton growing under different conditions (Klausmeier et al., 2004; Liefer et al., 2019; Inomura et al., 2020). Klausmeier et al. (2004) explained the N:P stoichiometry of phytoplankton in terms of three strategies. 'Survivalists' are rich in resource acquisition machinery (pigments and proteins) and have a high N:P ratio; 'generalists' have a balance between resource acquisition and growth machinery and an N:P ratio at or near Redfield proportions; 'bloomers' are adapted for exponential growth and have a high proportion of growth machinery, resulting in a low N:P ratio (Arrigo, 2015; Klausmeier et al., 2004). Based on this model, rapidly growing phytoplankton associated with ice nucleation at relatively warm temperatures have the characteristics of 'bloomers'. Rapidly growing phytoplankton have elevated cell quotas for P as cell processes required for rapid growth involve large quantities of nucleic acids. For example, growth requires P-rich ribosomes (composed of RNA and protein) for the synthesis of proteins. This concept is also expressed by ecologists in terms of the growth rate hypothesis (GRH), where the C:P and N:P ratio of fast-growing organisms is low due to the high concentrations of ribosomal RNA required for protein synthesis (Elser et al., 2000; Isanta-Navarro et al., 2022), though the application of the GRH to all phytoplankton has been challenged (Flynn et al., 2010).

Recent work with relatively pure samples of representative biomolecules from cells have shown significant differences in ice nucleation properties between compounds (Alsante et al., 2023; Steinke et al., 2020; Wolf et al., 2019). SSA collected from the MART tanks were a complex mixture of different biomolecules with different ice nucleation properties. These biomolecules were predominantly produced via photosynthesis and growth of the phytoplankton. Transformation of the phytoplankton organic matter by heterotrophic bacteria in the tank also played a role, as did background contamination of the ASW with organic matter. Determining how these components interact to affect immersion ice nucleation is a challenging and multifaceted problem. For example, concentration affects the immersion freezing temperature of the protein apoferritin, with higher concentrations resulting in freezing at warmer temperatures, due to protein aggregation (Cascajo-Castresana et al., 2020). The interactions of ice nucleating proteins with salts affect ice nucleation temperatures (Schwidetzky et al., 2021).

Our recent work shows that both DNA and RNA are moderately effective INPs, nucleating at a mean temperature of -20 ± 1 °C (Alsante et al., 2023). Given that DNA is reportedly enriched by a factor as high as 30,000 in artificially generated SSA using seawater from the North Atlantic Ocean (Rastelli et al., 2017), as well as direct metagenomic evidence of DNA in the atmosphere (Mayol et al., 2017; Lang-Yona et al., 2022), this represents a plausible marine INP. We did not measure the concentration of nucleic acids in the MART, but the GRH and 'bloomer' strategy would suggest that nucleic acids were a relatively large proportion of the organic matter during the early stage of the cultures, when they were growing at the highest rates and affected ice nucleation at the warmest temperatures. However, nucleic acids would have accumulated in the cultures after the period of relatively warm ice nucleation as the total biomass increased, suggesting that ice nucleation at warm temperatures was not driven by the concentration of nucleic acids.

Another potential source of INP in fast growing phytoplankton are proteins associated with growth. The measured N:C ratio of phytoplankton increases with growth rate (Inomura et al., 2020), indicating that fast growing phytoplankton are synthesising more proteins and nucleic acids, storing more nitrogen, or both. Recent studies have shown that proteins can be effective INP (Cascajo-Casresana et al., 2020; Daily et al., 2022; Lukas et al., 2021; Roeters et al., 2021; Scwhidetsky et al., 2021; Hartman et al., 2022). Proteins, like nucleic acids, would have accumulated in the cultures beyond the initial phase of fast growth rate as the phytoplankton biomass continued to increase. Therefore, the amount of protein in the water (and presumably the SSA generated from the water) was not a driver of the warm freezing temperatures observed during the early phase of growth. There are significant differences between proteins in their ability to act as INP (Cascajo-Casresana et al., 2020; Alsante et al., 2023). Some proteins are among the most efficient known INP. Examples of highly efficient protein INPs include InaZ from *Pseudomonas syringae*, which nucleates ice at temperatures as high as -2 °C (Maki et al., 1974; Roeters et al., 2021; Wex et al., 2015), and ribulose-1,5-bisphosphate carboxylase/oxygenase (RuBisCO) (Alsante et al., 2023). RuBisCO initiates immersion mode ice nucleation at -6.8°C, with complete freezing at -9°C, and a mean freezing temperature of -7.9 ± 0.3°C (Alsante et al., 2023). RuBisCO would have been abundant in the MART experiments as it is one of the most abundant proteins on Earth, which is ubiquitous in phytoplankton as it is essential to photosynthetic carbon fixation (Ellis et al., 1979; Raven 2013; Bar-On & Milo, 2019). In contrast, other proteins are significantly less efficient as INP; the mean immersion mode freezing temperature of alkaline phosphatase was -20.4 ± 1.3°C in the same study (Alsante et al., 2023). These findings suggest that it is the presence of specific proteins, rather than bulk protein concentration, that determine the ice nucleation efficiency of organic matter.

A representative phytoplankton cell is composed of 25–50% protein, 5–50% polysaccharide, 5–20% lipids, 3–20% pigments and 20% nucleic acids (Emerson & Hedges, 2008). While we did not measure lipids, total carbohydrates were used as an indicator of biomass in the culture of *T. weissflogii*. Carbohydrate concentration followed the same pattern as cell concentration, peaking after the measurement of most efficient ice nucleation. Carbohydrate breakdown became more significant as the culture aged, as indicated by measurements of glucosidase activity and the decrease in total carbohydrate concnetration after day 7. Wolf

et al. showed that both lipids and carbohydrates from the cyanobacterium *Prochlorococcus* were efficient deposition INP. Dreischmeier et al. (2017) showed that large polysaccharides derived from tree pollen acted as INP, whereas smaller carbohydrates had anti-freeze properties. It was not possible to determine whether carbohydrates and lipids acted as a source of INP in the MART tanks.

Differences in DOM concentration and composition could also play a role in explaining why ice nucleation temperature changes with growth rate and physiological status. The proportion of fixed carbon released as DOM depends on both taxon and physiological status (Thornton, 2014). Cell death also releases DOM into the surrounding water (Veldhuis et al., 2001; Thornton, 2014). The enrichment of SSA with organic matter has been related to stress and death in both field (O'Dowd et al., 2015) and laboratory experiments (Wang et al., 2015). In addition to physiological stress, physical disruption of cells affect the release of DOM and cell fragments from phytoplankton *in situ* through sloppy feeding by zooplankton (Møller et al., 2003; Møller, 2007) and viral infection (Bratbak et al., 1993; Vardi et al., 2012; Kranzler et al., 2019). Wolf et al. (2019) measured the ice nucleation biogenic material from *Prochlorococcus* cultures in the deposition mode, finding that small particles from lysed cultures were more effective INPs than large particles. Morison et al. (2019) found significant rates of grazer mortality and low rates of viral lysis at Station 4 in the Atlantic at the same time as our work, suggesting that these processes would have contributed cell fragments and DOM to SSA sampled *in situ*. Grazing and viral lysis did not occur in the MART experiments. However, SYTOX Green staining showed increased cell permeability during stationary and death phase of the *T. weissflogii* culture, indicating greater potential to leak DOM into the surrounding medium. Evidence for DOM in both MART experiments came from measurements of exopolymer particles and the presence of FDOM. FDOM in the MART showed that fluorescence peaks (peaks B and T; Coble, 1996) associated with proteins were high (Fig. S3b), both in terms of concentration and relative to other fluorescence peaks, during the exponential growth of *T. weissflogii* (Fig. 3). However, Peaks B and T were not elevated during exponential growth of *S. elongatus*. Based on our data, it was not possible to link ice nucleation activity with cell 'leakiness' and the measured components of DOM.

After the initial phase of rapid growth during the *S. elongatus* MART experiment there was a decline in biomass, as indicated by the halving of chl. *a* concentration between days 4 and 8. This was followed by a second period of growth, which did not correlate with relatively warm ice nucleation (Fig. 2). There are two possible explanations for this discrepancy. Firstly, the second period of growth (0.21 day$^{-1}$) was not rapid compared with growth rates during the early phase of the culture (1.34 day$^{-1}$). Secondly, the composition of the water had changed significantly since the initial phase of fast growth due to the death and physiological stress of *S. elongatus* between days 4 and 8. In addition to the declines in chl. *a* concentration and quantum yield of photosystem II, there was also an increase in the amount of dissolved organic matter in the MART, as indicated by an increase in FDOM concentration between measurements on days 1 and 6 (Fig. S3). While MART and mesocosm experiments often progress with one distinct peak (such as in the experiment with *T. weissflogii* (Fig. 3)) and compilation of growth curves in Lee et al. (2015), there are examples where there is more than one peak in phytoplankton biomass over the course of an experiment (Lee et al., 2015; McCluskey et al., 2017; Wang et al. 2015).

A separate blank MART experiment was run to determine the effect of the combination of salts and potential background contamination on the results. Ice nucleation measurements of these procedural blanks were higher than the homogeneous freezing temperature (-38 ºC; Kanji et al. 2017), but this was not surprising, and it is typical of procedural blank measurements (Alsante et al., 2023; Irish et al., 2017; Wilbourn et al., 2020). Two opposing factors affected the freezing temperature of the procedural blanks; salt depresses freezing temperature (Wilbourn et al. 2020; Perkins et al. 2020) and organic matter potentially act as an INP. The data suggest that background organic matter was more significant than salt depression in determining the ice nucleation temperature of phytoplankton-free medium. The procedural blanks suggest that the ASW contributed INP, both from the analytical grade salts

used to make the medium, and from the addition of nutrients (including vitamins and EDTA) to the growth medium. It is not possible to determine whether the INP activity of the medium was a direct effect of the organic matter background, or an indirect effect after the growth and transformation of organic matter by heterotrophic bacteria. These results show that ASW made with analytical grade salts offers no advantage over natural filtered seawater; both media will contain background organic matter contamination. The depression of freezing temperature by salts depends on the concentration and composition of salt, as well as the composition of potential INP interacting with the salt. Wilbourn et al. (2020) found that that decreasing the salinity of samples collected from the Atlantic Ocean from 30 g L$^{-1}$ NaCl to 3.75 g L$^{-1}$ increased heterogeneous ice nucleation freezing temperature by approximately 10 °C. However, interactions between salts and INP may be complex, Schwidetzky et al. (2021) found that some ions inhibited, while others facilitated, freezing in ice nucleating proteins from the bacterium *Pseudomonas syringae*. Further, the concentration of salts in solution at the time of freezing will not necessarily be the same as the original seawater generating the SSA. Wilbourn et al. (2020) proposed that SSA would activate as cloud condensation nuclei (CCN) before freezing, with the resulting water uptake diluting the salts in the sea spray by an estimated factor of approximately 27.          Number concentrations of INP were low ($< 2 \times 10^{-3}$ L$^{-1}$) during both MART experiments, even at temperature $< -20$ °C. Our concentrations were at the lower end of previous measurements from laboratory mesocosm and MART tanks (DeMott et al. 2016; McCluskey et al. 2017). Measurements of number concentrations of marine INP *in situ* are limited are limited in spatial and temporal coverage, and vary by orders of magnitude. A summary of the results from several recent campaigns in the Southern Ocean and found the INP concentrations at -20 °C were $10^{-4}$ to $10^{-2}$ per liter, with higher concentrations observed near the terrestrial influence of landmasses such as Australia (McFarquar et al., 2021). Welti et al. (2020) summarized published shipboard measurements of ice nucleation from the ocean, with a $10^{-4}$ to $10^{0}$ L$^{-1}$ range in INP concentration at -15 °C and a range in INP concentrations over 9 orders of magnitude ($10^{-4}$ to $10^{5}$ L$^{-1}$) over the temperature range -40 to -5 °C. These studies included some sites close to land, where continental sources were likely the major source of INP.

Field measurements (Wilbourn et al., 2020) corroborated our laboratory results; the fastest phytoplankton growth was associated with the warmest ice nucleation temperatures. According to Della Penna and Gaube (2019), the water sampled on Days 1-4 was sampled from within the same anticyclonic eddy. Similarly, Graff and Behrenfeld (2018) assumed that the same water mass was sampled for all 4 days at Station 4 in their study of phytoplankton photoacclimation. Following those studies, our interpretation of the field data assumes that the samples used to calculate net growth were from the same water mass.

Specifically, our growth rates (Fig. 5b) represent growth over the 24 hs leading up to the aerosol sampling; the growth rate for day 4 (Fig. 5b) was based on the differences in morning chl. *a* measurements (sampled between 8:00 and 9:00 local time) taken on days 3 and 4. Morison et al. (2019) measured grazing for days 1-3 on Station 4; grazing rates were insignificant on day 1 followed by significant on days 2 and 3 (0.26 to 0.44 day$^{-1}$; based on chl. *a*). Their observations show that *in situ* growth rates were significantly higher than we estimated. Unfortunately, Morison does not provide grow rates for Day 4, out of concern that day 4 was potentially biologically different as the ship had drifted from the center to the periphery of the eddy.

Fast growing *S. elongatus* facilitated ice nucleation at relatively warm temperatures in the laboratory and *Synechococcus* was a significant component of the phytoplankton biomass at Station 4 in the Atlantic (Wilbourn et al., 2020). *Synechococcus* had the fastest net growth rate (1.12 day$^{-1}$) between days 3 and 4, leading up to the relatively warm ice nucleation temperatures observed on day 4 at Station 4. The proportion of *Synechococcus* cells was relatively low; it contributed 9.8 and 14.5 % of total cell numbers counted with flow cytometry on days 3 and 4, respectively (Fig. 3a). Consequently, it was not possible to determine the role of organic matter from *Synechococcus* in affecting ice nucleation temperature. Other components of the phytoplankton also grew relatively quickly between days 3 and 4, including picoeukaryotes (0.68 day$^{-1}$), which were the most abundant components of the small phytoplankton (between 79.7 and 83.4 % of cells counted using flow cytometry). Wilbourn et al., (2020) also measured the

ice nucleation temperatures of *Synechococcus* isolated by flow cytometry on Day 4 at Station 4. There was no significant difference between the mean freezing temperature of *Synechococcus* and those of photosynthetic picoeukaryotes and photosynthetic nanoeukaryotes.

It should not be overlooked that heterotrophic microorganisms (bacteria and protists) were significant contributors to biomass at Station 4 (Bolaños et al., 2021). Bacteria play an important role in determining the fate of organic matter in the ocean (Kujawinksi, 2011; Hasenecz et al., 2020). Wang et al. (2015) proposed that the organic matter composition of SSA is affected not only by primary production, but also by the enzymatic degradation of organic matter by heterotrophic bacteria. Glucosidase activity was used as a proxy for bacterial activity in the MART culture of *T. weissflogii*; it increased by two orders of magnitude over the course of the experiment. This indicates that bacteria were playing a significant role in the remineralization and transformation of DOM in the MART tank, particularly as the culture aged during the death and stationary phases of culture. However, this had no observed influence on the either the concentration of aerosols or their ice-nucleating ability. Hill et al. (2023) found a positive correlation between the concentration of heterotrophic nanoflagellates and ice nucleating entities in a mini-MART experiment using natural seawater. Heterotrophic nanoflagellates were not present in the simplified systems grown in the MART, therefore our experiment does not account for their effects. Heterotrophic nanoflagellates were a component of the grazing community on the North Atlantic, though they were not counted (Morison et al., 2020).

Organic matter in the surface ocean can be divided into pools based on age as it exists on a continuum from labile to ultra-recalcitrant (Hansell 2013). From the perspective of the generation of SSA, we can consider a constant background pool of recalcitrant organic matter with a lifetime longer than the mixing time of the ocean, and relatively labile pools of organic matter that turnover quickly and are affected by recent biological processes, such as phytoplankton growth, in the surface ocean (Brooks and Thornton, 2018). There is a growing paradigm that the production of organic-rich primary SSA is driven by relatively labile organic matter produced recent processes, such as phytoplankton blooms, in the surface ocean (O'Dowd et al. 2004, 2015; Rinaldi et al. 2013; Sciare et al. 2009). However, there is no consensus as some studies indicate that the organic composition of primary marine aerosol is relatively constant, indicating that it determined by the relatively recalcitrant background pool of organic matter that is uncoupled from recent phytoplankton growth (Lewis et al. 2022; Quinn et al. 2014). INP represent a rare subset of atmospheric aerosol, with only 1 in $10^5$, or fewer, aerosol particles acting as INP (DeMott et al. 2010). This observation, coupled with the selective concentration of different pools of organic matter in the SML (Cunliffe et al., 2013; Thornton et al., 2016) and selectivity of the aerosolization process through bubble bursting (Aller et al., 2005, 2017; Rastelli et al. 2017), make it challenging to relate biological processes in the ocean with ice nucleation in the atmosphere.

Experiments with phytoplankton cultures can be controlled, and therefore are useful to understand the complex processes generating marine biogenic aerosol and INP (Hill et al., 2023; McCluskey et al., 2017, 2018; Prather et al. 2013; Wang et al. 2015). Recent work (Hill et al. 2023; McCluskey et al., 2017, 2018) suggest that the production of INP occurs during the collapse and decay of phytoplankton blooms during the later stages of a growth experiment. In contrast, our results that suggest that it is during exponential growth in the early stages of a growth experiment, when the most effective ice nuclei are produced. These findings are not mutually exclusive, INP may be significant at more than one stage of a phytoplankton bloom, in terms of abundance, relatively warm freezing temperature, or both. It is a consensus that ice nucleation at temperatures significantly warmer than homogeneous freezing is driven by organic matter in SSA. Therefore, processes that affect the composition and abundance of organic matter will affect both the abundance and properties of INP. Different phytoplankton have significantly different traits in terms of cell size, growth rates, and chemical composition (Beardall et al., 2009; Edwards et al., 2012; Ho et al., 2003; Quigg et al., 2003) and therefore the choice of taxa may affect the results. To date, too few taxa of phytoplankton have been grown in these types of experiment to determine taxa-specific patterns in INP production with growth. Our experiments were different from most other

MART or large tank experiments as we used artificial seawater as the growth medium; most other experiments have used natural seawater from the Pacific Ocean (Hill et al., 2023; McCluskey et al., 2017, 2018; Prather et al. 2013; Wang et al. 2015). This water was filtered through a mesh screen to remove animals and the largest protists, but it still contains natural assemblages of heterotrophic bacteria, protists and viruses. It may be that processes such as viral lysis, grazing, and the processing of phytoplankton organic matter by heterotrophic bacteria are important in determining the properties of SSA and the INP that they contain. Some of these processes were absent from our MART experiments. Differences between experiments may also arise from the research objectives of the researchers; Hill et al. (2023) focussed their experiment on the 'decay phase' of the phytoplankton bloom and McCluskey et al. (2017) did not make ice nucleation measurements between days 2 and 8 of a MART experiment.

4 ConclusionIn this study, we investigated whether the physiological status of phytoplankton, especially growth rates, can be linked to the properties of primary marine aerosol and ability to act as effective INPs. We initially anticipated that processes associated with the release of DOM and cell fragments into the water column associated with physiological stress, slow growth rates, and autocatalytic cell death would produce a larger number of more efficient INPs. Bloom collapse has been proposed as a major source of organic matter in SSA (O'Dowd et al., 2015). McCluskey et al., (2017) found an increase in the production of INPs, active between -25 and -15 ºC, following the peak in chlorophyll concentration in a mesocosm experiment with natural seawater. They explained this consistent population of INPs as resulting from the collapse and decay of the phytoplankton blooms during the mesocosm experiments (McCluskey et al., 2017). Our data indicate the opposite; ice nucleation at relatively warm temperatures was associated with fast growth, healthy photosynthetic machinery (indicated by high values of $\Phi_{PSII}$), and relatively few dying or leaking cells (indicated by SYTOX green staining).

Significantly, our results are the first to show that fast growing phytoplankton are a source of INPs that catalyse freezing at relatively warm temperatures. Mean freezing temperatures during the early growth phase of the MART cultures were > -24 ºC, which was warmer than the mean freezing temperature of the procedural blanks (-34.9 for ASW and -31.3 ºC for ASW+L1 nutrients). Insights into the relevant chemical properties of fast-growing cells will come from considering how phytoplankton allocate resources, with rapid growth requiring a relatively high cellular content of nucleic acids and specific proteins, compared with slow growth. Similar results were observed in two contrasting taxa of laboratory-grown phytoplankton and a taxonomically mixed population of phytoplankton in the ocean, indicating that the coupling between fast growth and efficient INPs is widespread phenomenon and not associated with any one taxonomic group. These results may have significant implications for the prediction of mixed phase cloud formation, life cycle, and precipitation on regional scales as rapidly growing phytoplankton form blooms over large areas of the ocean. Blooms such as the annual spring bloom in the North Atlantic Ocean may be a temporally and spatially predictable source of relatively active marine INPs in SSA.

**Data availability**

Data will become available in the Texas Data Repository (https://dataverse.tdl.org/) on publication.

**Supplement**

The supplement related to this article is available online.

**Author contributions**

SDB and DCOT designed the experiments with input from EKW, JM and ANA. EKW, JM, ANA, AW, and KM conducted the marine aerosol reference tank (MART) experiments, and analysed samples and data, in consultation with SDB and DCOT. GGB

analysed samples and processed the data for the CDOM analysis. DCOT wrote the manuscript with assistance from SDB and ANA, and editorial input from EKW, JM, and GGB.


**Competing interests**

The authors declare no competing interests.

**Acknowledgements**

We thank Joseph Niehaus for his significant contribution to the construction and testing of the marine aerosol reference tank. We thank the Captain and crew of the *R.V. Atlantis*, and the NAAMES science team, for making the collection of field samples possible.

**Financial Support**

SDB and DCOT were supported by the National Science Foundation (United States) under Grant No. AGS-1539881. Any opinions,

findings, and conclusions or recommendations expressed in this material are those of the authors and do not necessarily reflect the views of the National Science Foundation. Funding for field sample collection was provided by NASA Earth Venture Suborbital-2 (EVS-2) Award #NNX15AE68G.

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
