# Peer review of "Production of aerosol containing ice nucleating particles (INPs) by fast growing phytoplankton"

_Atmospheric Chemistry and Physics, 2022_

## Author Comment (AC2)

**RCP-1**

Overview:

The authors describe a detailed investigation of sea spray aerosol (SSA) as source of ice nucleating particles (INPs). Using an aquarium tank, two essential marine microbes, namely the cyanobacterium *Synechococcus elongatus* and the diatom *Thalassiosira weissflogii,* are cultivated and SSA is generated artificially. The aerosol was sampled using an impactor and analyzed for ice nucleation activity with a microscopic setup. Furthermore, the growth and change of phytoplankton cells was monitored using a comprehensive bioanalytical workflow. The authors try to correlate ice nucleation activity with growth status and compare results from the laboratory experiments with data collected from a field campaign in the North Atlantic Ocean.

Surprisingly, the findings of the paper suggest that ice nucleation activity does not scale with the total mass of microbes, but rather increases at the early growth period. After 3 days of incubation, the growth rate for both cultures drops, and the nucleation temperature of the sampled aerosol decreases significantly (Figure 2 and Figure 3). However, the cell count stabilizes before the death phase begins around day 10 (see Figure 3a). This observation suggests that the total amount of cells is not responsible for the ice nucleation activity. It is rather a qualitative variable (e.g. composition of DOM) that triggers the high freezing temperatures in the first days of the experiments. These findings contrast with other papers suggesting high INP concentrations peak after the blooming period (McCluskey et al., 2017).

Overall, the authors introduce the research to a broad field of readers very well, describe the experimental procedure in great detail and provide a meaningful discussion and conclusion. The papers should be considered for publication after addressing the following questions and revisions:

 **Reviewer's Comment:**

General questions and comments:

- The background of the freezing experiments (blank) varies between -34.9 and -31.3 °C, for artificial salt water, and artificial salt water plus nutrients, respectively. Can the authors comment on the cause of these freezing temperatures? Is the higher measured background attributed to the nutrients in the water? Can the nutrients induce heterogeneous ice nucleation?
- How does sea salt influence the freezing temperatures of the aerosols (see e.g. Perkins et al., 2020)? Is a freezing point depression possible within the experiments?

**Authors' response:** The reviewer is correct that the blanks froze at relatively warm temperatures, given that the homogeneous freezing of pure water droplets in the atmosphere is -38 °C (e.g. Kanji et al. 2017). To address their second question, yes, there is salt in the aerosols and salt is well known to depress freezing temperatures (see below).

Note that the blanks were *procedural* blanks, not blanks run with pure water, which are included in the study specifically because the procedure entails a number of steps in which potential contamination could be introduced. Two opposing factors affected the freezing temperature of the collected aerosol, salt depresses freezing temperature (Wilbourn et al. 2020; Perkins et al. 2020) and organic matter potentially act as an INP. The data suggest that background organic matter was more significant factor than salt depression in determining the ice nucleation temperature of phytoplankton-free medium.

To elaborate, the blanks were processed in the same way as the actual samples; they were generated as aerosol from the MART tank and collected onto aluminum foil using the same procedure as the culture samples (see revised section 2.4 for an explanation). The procedural blanks were collected from the blank MART experiment (i.e. no phytoplankton were added to the MART; see section 2.4). The two blanks represent different stages of the blank MART experiment. The closed black circles (Fig. 2a and 3a) show the mean freezing temperature of artificial seawater (ASW) blanks, without the addition of L1 nutrients. This medium contained the inorganic salts that make up sea water. The open white circles (Fig. 2a and 3b) show the mean freezing temperature of blanks after the addition of L1 nutrients. The L1 nutrients added organics to the medium in the form of vitamins in nM concentrations ($B_{12}$ (1 nM), biotin (4 nM), and thiamine (297 nM). A second source of organics was added in the form of 15 µM EDTA (ethylenediaminetetraacetic acid [$CH_2N(CH_2CO_2H)_2]_2$), a chelator for the trace metals in the medium. The L1 nutrient addition also included the addition of inorganic macronutrients (60 µM $NaNO_3$, 20 µM $NaH_2PO_4$, and 60 µM $Na_2SiO_3$.) and trace metals (Harrison et al. 1980, Berges et al. 2021).

The procedural blanks of aerosolized ASW had a mean (± SD) freezing temperature of -34.9 ± 0.9 °C, which increased to -31.3 ± 0.9 °C for ASW + L1 nutrients. The addition of the organic matter with the L1 nutrients in the ASW + L1 medium likely accounts for the observed increase in freezing temperature, by directly catalyzing homogeneous freezing, or by providing substrate for the growth of heterotrophic bacteria within the MART, which were INP or transformed organic matter into INP.

The artificial sea water (ASW) (Harrison et al. 1980, Berges et al. 2021) used in these experiments is a complex mixture of salts. The full medium contains ultra-high purity water and 25 different commercially sourced chemicals. These vary in concentration from 363 mM (NaCl) to 1 nM ($Na_2SeO_3$) in the final medium. High-purity, analytical-grade, chemicals were used to prepare the growth medium for the MART to reduce potential for contamination. However, any contamination of these salts is a potential source of background INP, especially given the large mass of salts added to the medium. The relatively warm freezing temperatures of the ASW procedure suggests background contamination with organic matter in the salts. The following text has been added to section 2.1 (lines 102-106):

"The ASW was made with high purity analytical grade salts. Nevertheless, the large mass of salts in artificial seawater represents a source of potential contamination. To reduce this possibility, sodium chloride was combusted for 6 hours at 500 °C to remove organic contamination. This precaution was taken given the large amount of sodium chloride in the medium (21.19 g $L^{-1}$). For other salts, notably the hydrated salts, combustion to remove potential organic contamination was not an option as it would have changed the composition and solubility of the salts."

In conclusion, we would not expect procedural blanks to freeze at -38 °C as they were not pure water. The difference between the two types of procedural blanks suggest that the organic matter (EDTA and vitamins) added to the medium enhanced ice nucleation enhanced ice nucleation directly or indirectly by supporting the growth of bacteria. It is likely that background organic matter contributed to ice nucleation in the seawater salts. Salts are also known to depress ice nucleation. Therefore, the freezing temperature of the procedural blanks represent complex interactions between the salt and organic matter in the sea spray collected in the aerosol samples.

The following has been added to the discussion (section 4; lines 601-628):

"A separate blank MART experiment was run to determine the effect of the combination of salts and potential background contamination on the results. Ice nucleation measurements of these procedural blanks were higher than the homogeneous freezing temperature (-38 ºC; Kanji et al. 2017), but this was not surprising and it typical of procedural blanks in measurements (Irish et al. 2017; Alsante et al. 2023). Two opposing factors affected the freezing temperature of the collected aerosol, salt depresses freezing temperature (Wilbourn et al. 2020; Perkins et al. 2020) and organic matter potentially acts as an INP. The data suggest that background organic matter was more significant than salt depression in determining the ice nucleation temperature of phytoplankton-free medium. The freezing temperature of the procedural blanks were warmer than homogeneous freezing. The procedural blanks suggest that the medium contributed INP, both from the analytical grade salts used for the medium, and from the addition of nutrients (including vitamins and EDTA) to the growth medium. It is not possible to determine whether the INP activity of the medium was a direct effect of the organic matter background, or an indirect effect after the growth and transformation of organic matter by heterotrophic bacteria. These results show that ASW made with analytical grade salts offers no advantage over natural filtered seawater; both media will contain background organic matter contamination. The depression of freezing temperature by salts depends on the concentration and composition of salt, as well as the composition of potential INP interacting with the salt. Wilbourn et al. (2020) found that decreasing the salinity of samples collected from the Atlantic Ocean from 30 g L$^{-1}$ NaCl to 3.75 g L$^{-1}$ increased heterogeneous freezing temperature by approximately 10 °C. However, interactions between salts and INP may be complex, Schwidetzky et al. (2021) found that some ions inhibited, while others facilitated, freezing in ice nucleating proteins from the bacterium *Pseudomonas syringae*. Further, the concentration of salts in solution at the time of freezing will not necessarily be the same as the original seawater generating the SSA. Wilbourn et al. (2020) proposed that SSA would activate as cloud condensation nuclei (CCN) before freezing due to immersion INP, with the resulting water uptake diluting the salts in the sea spray by an estimated factor of 27."

**Reviewer's Comment:** A main concern arises after considering the high background and the frozen fractions of day 4 to 14 for *Synechococcus elongatus* and day 9 for *Thalassiosira weissflogii.* To my understanding these days show freezing activity close or even below the blank level. Still the authors discuss the freezing temperatures as results of the microbiological activity. Are the freezing events caused by microbes or do the freezing temperatures just drop to the background level? The authors should comment on that question and consider rephrasing corresponding explanations given in the manuscript. Furthermore, the blank freezing curves

should be included in Figure 2c and Figure 3c. In case of any doubt, I suggest performing a statistical test to prove significant differences between the blank freezing temperatures and samples.

**Authors' response:** We thank the Reviewer for this suggestion. As suggested, fraction frozen curves for both types of procedural blanks (i.e. ASW and ASW+L1) from the blank MART experiment have been added to Figures 2c and 3c. These are a useful addition as they emphasize how the blanks relate to the experiments.

The text has been changed (see lines 376 - 380 and lines 388- 389) to emphasize that the freezing temperature of many of the samples cannot be considered different to the procedural blanks:

"Plots of fraction frozen against freezing temperature show that, except for warm freezing temperatures on days 2 and 3, the freezing temperatures of aerosol from the *Synechococcus* culture were between those of the procedural blanks; generally warmer than ASW on its own, but colder than ASW+L1. Therefore, except for days 2 and 3, there was no difference between the freezing temperature of SSA containing phytoplankton organic matter and the procedural blanks."

"Fraction frozen curves showed that freezing occurred at temperatures significantly warmer than the procedural blanks at the beginning of the experiment (days 2 and 3) and at the end of the experiment (days 16 and 29). "

**Reviewer's Comment:** The results from the MART experiments and the field data indicate that the ice nucleation activity does not only scale with biomass but is rather a complex function of the composition of organic matter involved in the system as seen for terrestrial INP populations (Steinke et al., 2020). The authors comment on that finding and give possible ice nucleating particles as explanations (e.g. DNA or proteins). How would the DNA and protein content change during the early stage of the growth phase? Is the overall expression of DNA, proteins etc. or the expression of specific molecules responsible for ice nucleation activity? Are expected freezing temperatures for proteins generally higher (e.g. Schwidetzky et al.,2021)? Are polysaccharides possible ice nucleators causing the freezing behaviour in the study (e.g. Dreischmeier et al., 2017)?

**Authors' response:** Discussion of the potential role of phytoplankton composition on ice nucleation has been revised and significantly expanded throughout the discussion (see lines 512 – 572)

**Reviewer's Comment:** Extent the discussion about the cause of differences to other studies (McCluskey et al., 2017).

**Authors' response:** Discussion of the difference in our work from previous studies has been extended (lines 676 – 695). We have made major revisions to the discussion in response to the comments from both Reviewers.

**Specific comments:**

**Reviewer's Comment:** Line 20: Consider rephrasing to "Ice nucleation occurred at colder temperatures (blank level) …"

**Authors' response:** Line 20 to 22 changed to: "Ice nucleation occurred at colder temperatures (-28.5 °C and below), which were not different from the freezing temperatures of procedural blanks, when the cultures were in the stationary or death phases of growth."

**Reviewer's Comment:** Line 24: Introduce the reader to the field campaign to avoid confusion (e.g. "We conducted a field measurement in the North Atlantic Ocean to compare the laboratory study with environmental data." or something similar).

**Authors' response:** Lines 25 to 27 changed to: "For comparison, field measurements in the North Atlantic Ocean showed that high net growth rates of natural phytoplankton assemblages were associated with marine aerosol that acted as effective immersion INPs at relatively warm temperatures."

**Reviewer's Comment:** Line 94: Is 27°C to 28°C comparable with real temperatures in the Northern Atlantic Ocean? If not, why was such a high temperature chosen in your experiment?

**Authors' response:** The temperature was chosen for practical reasons in maintaining a stable culture temperature given the balance between heating from the lighting and instrumentation in the room and the available cooling systems. The experiment was not designed to be analogous to the North Atlantic as it was a relatively simple system in which model phytoplankton taxa were used. Our model diatom (*Thalassiosira weissflogii* CCMP 1051) was originally isolated from the North Pacific Ocean. We used this taxon as Thornton has used it in multiple published experiments (Chen & Thornton 2015; Chen et al. 2021; Rzadkowolski & Thornton 2012; Thornton & Chen 2017). *Synechococcus elongatus* (CCMP 1379) was originally isolated from the South Pacific. It was also used as a representative and well characterized taxon, rather than a direct representative of the Atlantic. However, both *Thalassiosira* and *Synechococcus* are found in the Atlantic Ocean.

**Reviewer's Comment:** Line 112: Was the waterfall only 3.56 x 10^-3 m high?

**Authors' response:** The waterfall was $3.56 \times 10^{-1}$ m high – the text has been corrected.

**Reviewer's Comment:** Figure 1: Consider including the stir bars and the LEDs in the schematic drawing.

**Authors' response:** These have been added to Figure 1, as requested.

**Reviewer's Comment:** Figure 2b: Why are the error bars only shown for 2 samples?

**Authors' response:** There were error bars on all data points on Figures 2b and 3b; In most cases the error bars (standard deviations) were small enough to be covered by the data point. In those cases where there were visible error bars it was because they were large enough to be seen outside of the data points. The following phrase has been added to the text:

"The standard deviations of measurements of both $\varphi_{PSII}$ and chlorophyll $a$ were smaller than the data points in most measurements in…"

This statement was added for both Fig 2b (lines 364-365) and Fig. 3b (Lines 384-385).

**Reviewer's Comment:** Figure 4: Consider excluding lower freezing samples and matching Figure a and b to one figure for a better comparison. Furthermore, cumulative number concentrations of INPs are often plotted on a logarithmic scale (see e.g. DeMott et al., 2015). A literature reference could be included to support if the data represent other field measurements of INPs.

**Authors' response:** The scales on the two graphs in Figure 4 have been matched to ensure that the graphs are directly comparable. The axes of Fig 4b were changed to match those of Fig 4a. Note that Day 7 has been removed from Fig. 7b. This was done in response to reviewer 2's comments about the aerosol measurements and the decision to remove day 7 aerosol number concentration from Figure S1.

The reviewer is correct, these types of data are often plotted on a logarithmic scale (e.g. Wilson et al. 2015; DeMott et al. 2015; 2016). Some of these previous papers are comparing data collected in very different environments with large variations in underlying conditions and a logarithmic scale is necessary to visualize and compare the data over the entire range of values. As the MART data did not vary over orders of magnitude, we decided that a log scale was not necessary or helpful.

**Reviewer's Comment:** Line 322: Write -34.9 instead of 34.9°C.

**Authors' response:** The freezing temperatures on line 345 have been corrected.

References:

McCluskey, C. S., Hill, T. C. J., Malfatti, F., Sultana, C. M., Lee, C., Santander, M. V., Beall, C. M., Moore, K. A., Cornwell, G.C., Collins, D. B., Prather, K. A., Jayarathne, T., Stone, E. A., Azam, F., Kreidenweis, S. M., and DeMott, P. J.: A dynamic link between ice nucleating particles released in nascent sea spray aerosol and oceanic biological activity during two mesocosm experiments, J. Atmos. Sci., 2017.

Perkins RJ, Vazquez de Vasquez MG, Beasley EE, Hill TC, Stone EA, Allen HC, DeMott PJ.: Relating Structure and Ice Nucleation of Mixed Surfactant Systems Relevant to Sea Spray Aerosol. The Journal of Physical Chemistry A., 2020.

Steinke I, Hiranuma N, Funk R, Höhler K, Tüllmann N, Umo NS, Weidler PG, Möhler O, Leisner T.: Complex plant-derived organic aerosol as ice-nucleating particles–more than the sums of their parts?. Atmospheric Chemistry and Physics, 2020.

Schwidetzky R, Lukas M, YazdanYar A, Kunert AT, Pöschl U, Domke KF, Fröhlich-Nowoisky J, Bonn M, Koop T, Nagata Y, Meister K.: Specific Ion–Protein Interactions Influence Bacterial Ice Nucleation. Chemistry–A European Journal, 2021.

Dreischmeier K, Budke C, Wiehemeier L, Kottke T, Koop T.: Boreal pollen contain ice-nucleating as well as ice-binding 'antifreeze polysaccharides. Scientific reports, 2017.

**RCP-2**

Thornton et al., have described findings from two marine aerosol reference tank experiments they performed to determine the ice nucleation behavior of aerosol generated during the rapid growth of a cyanobacterium (*Synechococcus elongatus*) and a diatom (*Thalassiosira weissflogii*). The authors also compare their laboratory studies with field measurements from the North Atlantic. This study is motivated by a need to characterize marine INPs. The main conclusion of this paper is "Significantly, our results are the first to show that fast growing phytoplankton are a source of INPs that catalyze freezing at relatively warm temperatures". This conclusion is not supported by the experimental or field data, as described below. While the authors carefully explain the methods, there are a number of concerns with their experimental design that must be addressed. My recommendation is to reject this manuscript due to the number of major concerns regarding the experimental design (including likely background INP, unrealistic INP number concentrations, and poorly constrained aerosol measurements), a lack of evidence to support their conclusions, and a major over-generalization of their results.

**Major Comments:**

**Reviewer's Comment: INP number concentrations:** The authors do not address how their measurements compare to observations or previous laboratory studies (for which there are many). This is concerning given the large discrepancy between their reported INP number concentrations and the existing literature. INP number concentrations over marine regions are usually greater than $0.001 L^{-1}$ at -20C from mesocosm experiments (McCluskey et al., 2017, Mitts et al., 2021) and observations of aerosol sampled from the north Atlantic (McCluskey et al., 2018), aerosol sampled over the Southern Ocean (McFarquhar et al., 2021), or any open ocean region (Welti et al., 2020). The INP number concentrations reported in this manuscript are significantly lower, with all but 4 samples (of 15 total) containing undetectable INP number concentrations at -20C. It is possible the authors have made a simple calculation error, but this must be addressed before this paper can be consider for publication.

**Authors' response:**

We disagree with the reviewer's assessment that values in the literature are generally higher. The best place to obtain measurements of INP concentration in a real world setting is through shipboard measurements. As summarized by McFarquhar (2021), variability in INP concentrations sampled in the Southern Ocean during 2 shipboard projects (Capricorn I and II, Marcus, and TAN1502) indicated that INP concentrations in the open ocean were very low, $10^{-4}$ to $10^{-3}$ INPs per liter. Broadly speaking our data is in good agreement with these measurements (though this is by no means an apples-to-apples comparison.) The study also notes higher concentrations in coastal waters and near continents which is consistent with the higher concentrations observed from both field sampling from land sites near the ocean and with MART tank studies which use coastal waters in the tank (McClusky et al., 2017).

To elaborate, the Reviewer is incorrect to say that there is a "discrepancy" between the results reported here and the literature. There are differences between the observations here and other studies, of course, but the Reviewer implies that our results should match other work in

mesocosms, which is a gross oversimplification. It is wrong for the Reviewer to conclude anything is "usually greater" based on such a limited number of studies. To address, "INP number concentrations over marine regions are usually greater than 0.001 L$^{-1}$ at -20C from mesocosm experiments (McCluskey et al., 2017, Mitts et al., 2021)". Here the review cites only 2 papers, representing only 4 mesocosm experiments.

In their summary of field measurements, Welti et al. (2020) reports orders of magnitude ranges in INP concentration in a given ocean, indicating considerable 'real world' variation in INP concentrations. On their world ocean map (Fig. 1), there is a 4 order of magnitude range in INP concentration at -15 ºC and a9 order of magnitude range in INP concentrations when temperatures from -5 to -40 ºC are considered (Figure 4). It is also important to note that the values in Welti et al. (2020) are in different units than ours - their data for INP number concentrations are per m$^3$, and therefore should be divided by 1,000 for direct comparison to our results.

Of course, there are caveats to field studies as well, including that many organism types will be present all at once and that different taxa of phytoplankton exhibit different growth rates and patterns of growth. Wilbourn et al. addressed the issue of organism type by using cell sorting flow cytometry to separate different groups of phytoplankton from natural sea water, which were then used in ice nucleation measurements (Wilbourn et al, 2020).

Finally, we note that McCluskey et al. (2018) was sampled at Mace Head, which is not comparable as it is a coastal site influenced by continental aerosol. As discussed in Wilbourn et al. (2020), while ground based measurements have sometimes been referred to as "marine" INP concentrations measurements collected are spuriously high comparisons to shipboard measurements and should be labeled something else.  In McCluskey et al., (2018), the measurements collected from the direction of the ocean had to pass over the coastal zone containing notoriously high concentrations of kelp, which has also been suggested as a source of aerosol. For these reasons, we prefer to leave McCluskey et al. (2018) out of this discussion. Finally, even over the open ocean, transported continental aerosol can be a major source of INP. For this reason, composition analysis has been employed as a tracer of continentally influenced air masses and to identify the rarer cases of clean marine air (Saliba et al., 2020).

**Now that we have addressed issues in the Reviewer's comment, we move on to the heart of the comment, INP concentration.** We will include a brief summary of previous INP observations and how they are like and unlike the results reported here. This will strengthen the manuscript.

First, while the MART tank is touted as the best representation of aerosols of the composition and size distribution of naturally occurring sea spray, one caveat is that there is no ideal method to produce realistic marine aerosol concentrations in a laboratory setting. This is very hard to overcome, given that aerosol concentration in the open ocean is strongly influenced by wind speed (Saliba et al.2019), which is not replicated in the MART tank technique. Despite the

inherent uncertainties, there are some reports of INP concentrations measured in MART tanks. Like our own study, these are meritorious in their determination of relative changes in INP concentration.

For example, Mitts et al. (2021) report that total INP are present in a mesocosm in concentrations an order of magnitude higher than submicron INPs (at -20 C, these are ~$5x10^{-2}$ per liter and $1-10$ $x10^{-3}$ per liter, respectively. By comparison, our samples were also **submicron measurements** collected on the PIXE L1 impaction stage (0.06-1 μm aerodynamic diameter). At $1x10^{-3}$ per liter at -20 C, our data overlaps the lower concentrations of the submicron data observed by Mitts. In another MART tank study, 2 mesocosm runs were conducted using coastal seawater from California. For comparison, at -20C in the first run concentrations were $10^{-3}$ to $10^{-2}$ INPs per liter (according to McClusky et al. (2017) Figure 1) and nearly an order of magnitude higher (up to $10^{-1}$ INPs per liter) in the second run. The reasons are uncertain since, as McClusky stated, "However, the interpretation of this trend is limited because of the inability to measure from the MART system during the growing phase". However, one possible explanation is differences in composition of the biological constituents between the 2 runs and another is differences in the growth phases of the organisms sampled.

The text now reads (Lines 620-628):

"Number concentrations of INP were low (< $2 x 10^{-3}$ $L^{-1}$) during both MART experiments, even at temperature < -20 °C. Our concentrations were at the lower end of previous measurements from laboratory mesocosm and MART tanks (DeMott et al. 2016; McCluskey et al. 2017). Measurements of number concentrations of marine INP *in situ* are limited are limited in spatial and temporal coverage, and vary by orders of magnitude. A summary of the results from several recent campaigns in the Southern Ocean and found the INP concentrations at -20 °C were $10^{-4}$ to $10^{-2}$ per liter, with higher concentrations observed near the terrestrial influence of landmasses such as Australia (McFarquar et al., 2021). Welti et al. (2020) summarized published shipboard measurements of ice nucleation from the ocean, with a $10^{-4}$ to $10^{0}$ $L^{-1}$ range in INP concentration at -15 °C and a range in INP concentrations over 9 orders of magnitude ($10^{-4}$ to $10^{5}$ $L^{-1}$) over the temperature range -40 to -5 °C. These studies included some sites close to land, where continental sources were likely the major source of INP."

**Reviewer's Comment: Experimental INP background:** The authors report that they have evidence for INPs emissions associated with the rapid growth of phytoplankton based on these two MART experiments. In addition to the discrepancies between the literature and the reported INP number concentrations in this experiment, I am concerned that the authors did not test the growth medium used to grow the phytoplankton culture. Given that INPs are rare in the atmosphere, but difficult to avoid in laboratory experiments, this is a major concern. In fact, a study by Fröhlich-Nowoisky et al. (2015) found that the initial medium choice was ice nucleation active at temperature lower than -12C ("We originally intended to grow the isolates on malt extract agar. However, since the available product was found to contain some IN (active

at $-12$ °C), an approximate equivalent using IN-free ingredients (tested to $-18$ °C) was constructed.") This is a major shortfall of the study, as the initial signal in the first 2-3 days may very well be a signal from an initial emission of INPs from the medium used that fell off after a few days. If the authors chose to resubmit this study, they need to show tests of the medium used in their experiment.

**Authors' response:** The Reviewer seems to have missed basic details of how our experiment was designed and how we conducted many blanks to ensure we did not misinterpret observations of freezing. The Review is correct that the medium is important, and for that reason we ran a control 'blank' MART experiment with medium alone to test the impact of the medium on ice nucleation. This experiment was described in the original manuscript (section 2.4, which has been revised for clarity). The addition of organics with the salt used to make up the medium was a potential issue (one which we addressed by using analytical grade salts and removing organic matter from NaCl by heating it in a combustion oven (see methods)). In short, this so-called " major shortfall of the study" is not a shortfall at all.

Since the Reviewer missed the discussion in the text, we expand on the text as described at the end of this comment. First, we elaborate on the issue here**. See also the discussion of blanks in response to Reviewer 1 above.**

In any system suitable for phytoplankton growth (MART tank or otherwise) there will be background organic matter in the medium, whether natural seawater or artificial seawater. Background organic matter is a known issue in experiments run with natural seawater. After filtration through a GF/F filter, the surface open ocean contains dissolved organic carbon (DOC) concentrations of 34 to 80 µmol kg$^{-1}$, much of which is relatively recalcitrant (Hansell et al. 2009). The effect of this background DOC is rarely considered, though it has been seen as the driver for relatively constant SSA properties, where phytoplankton growth is decoupled from the composition of primary aerosol over the ocean (Lewis et al. 2022; Quinn et al. 2014) (see lines 663-675 in the manuscript). Any experimental system where nutrients are added will include the potential for unwanted background organic contamination and the deliberate addition of organic matter via vitamins and EDTA (e.g. see table S2 of Lee et al. 2015).

It is unfair to equivocate our experiment with that of Fröhlich-Nowoisky et al. (2015). They were growing a terrestrial fungus in the laboratory on an artificial media, which is a very different situation from growing phytoplankton in artificial seawater. As a heterotroph, a soil fungus requires organic matter to grow. Fortunately, phytoplankton obtain their organic carbon from photosynthesis. This minimizes the addition of organic matter to essential organic compounds that the phytoplankton may not be able to synthesize themselves (i.e. some vitamins and EDTA to chelate trace metals). The total amount of organic matter that we added to our MART experiments was 0.35 g (0.006 g L$^{-1}$ of artificial sea water). Adding the organic ingredients of Fröhlich-Nowoisky et al. (2015) to our experiment would have meant adding 1,153 g of organic matter to the MART! Much of the organic matter that Fröhlich-Nowoisky et al. (2015) added to their medium was in the form of peptone (3 g L$^{-1}$), which is a poorly defined mixture of soluble

proteins manufactured from the partial hydrolysis of protein from a variety of sources (e.g. cattle, milk, soy etc.). Given the importance of proteins as biogenic INP, it is fortunate that we did not have to add sources of protein to our experiments. We clearly cite the references for artificial seawater in our text (Berges et al., 2001; Harrison et al., 1980; Guillard and Hargraves, 1993). These media are widely used to culture phytoplankton and have a long history. Readers can consult these papers for the details on how these media are made and their exact composition.

Samples from the control MART containing only growth medium were generated using the same methods as actual samples and these procedural blanks were measured for ice nucleation to determine the effect of the medium on our measurements. Two types of procedural blank were taken – one representing the ASW medium on its own (i.e. a solution of inorganic salts) and a second representing the ASW medium after the addition of L1 nutrients (trace metals and vitamins) and the inorganic nutrients that we added (nitrate, phosphate, silicate) at significantly lower concentrations than the original L1 medium (see section 2.1 for concentrations).

As presented in the manuscript, the procedural blanks that were tested for ice nucleation activity did catalyze the freezing of water at temperatures significantly above homogeneous freezing and at temperatures that were not significantly different from many of the samples taken from the experimental MARTs. As presented in the discussion, we saw warmer ice nucleation temperatures in the control MART after the addition of nutrients, either due to the direct effect of the organics added to the tank, or as an indirect effect of bacteria processing that small amount of organic matter. The presentation of the results and discussion has been improved to ensure that these effects are clear to readers.

We have now expanded the discussion of the medium experiment with edits to the methods (section 2.4), results section (e.g. lines 376-380 and lines 388-3389) and discussion (Lines 601-619). We refer the reviewer to these significant revisions and hope that are use of procedural blanks is clearer.

**Reviewer's Comment: Inconsistent relationship between INPs and exponential growth rate:** The authors claim in the abstract and conclusions that "Ice nucleation at warmer temperatures was associated with relatively high values of the maximum quantum yield of photosystem II (PSII), an indicator of the physiological status of phytoplankton." Where, the quantum yield of photosystem II is their indicator for exponential growth. In the *S. elongatus* experiment, there are two peaks in the quantum yield of photosystem (days 5 and 11), yet the INP activity is undetectable at temperatures above -19C for all days except for Day 3. In the *T. weissflogii* experiment, the quantum yield of photosystem II peaks on day 1 slowly declines through the 15 days; there are only 6 days of INP measurements (missing data unexplained) and all samples are inactive at temperatures above -19C. The field observation growth rates do not exceed 0.6 day$^{-1}$, which is significantly lower than the elevated growth rates of the two MART experiments (1.3 and 3 day$^{-1}$). Do the MART experiment growth rates

represent anything of the natural ocean? All in all, these inconsistent and weakly related data do not support their conclusions.

**Authors' response:** The reviewer is incorrect - the quantum yield of photosystem II ($\phi_{PSII}$) is not a measure of phytoplankton growth and was not used as an indicator of growth. Growth was determined from changes in proxies for biomass; namely cell counts, chlorophyll, and bulk carbohydrate. We measured the quantum yield of photosystem II as it is a measure of the physiological status of the phytoplankton (lines 209-220) and therefore it is not directly coupled to growth. It basically tells us how 'healthy' the phytoplankton are. Specifically, the quantum yield of photosystem II tells us what proportion of captured light is channeled into photosynthetic photochemistry rather than being lost as heat or fluorescence.

It would be inappropriate to use quantum yield of photosystem II as a measure of growth. It is sometimes and inappropriately used as an indicator of photosynthesis rates, but it is not a direct measure of photosynthesis as it relates to the capture of light energy (electron transfer) through the 'front end' of photosynthesis rather than carbon fixation at the 'back end'.

Therefore, the reviewer should not view decreases in quantum yield of photosystem II as a decrease in growth rate or that high values of quantum yield of photosystem II are a measure of exponential growth. Quantum yield of photosystem II may be uncoupled from growth, though as photosynthesis is essential to growth, one would expect that low quantum yields of photosystem II would eventually lead to less resources available for growth.

**Secondly, regarding Growth rates** – the reviewer is correct that there is a difference between 'high' growth rates between the diatom (*Thalassiosira*) and the cyanobacterium (*Synechococcus*), and the field samples. Growth rates are not directly comparable between taxa and different ranges of growth rates are found in different taxa of phytoplankton – so a high growth rate for one group of phytoplankton is not a high growth rate for another group. The growth rate of *Thalassiosira* (Fig. 3a) was very high, but relatively high growth rates are an established trait of diatoms (Inomura et al. 2023) and we would not expect 'high' growth rates in diatoms to be the same as cyanobacteria. This is not a flaw in the experimental design, but rather reflects fundamental differences in the biology of different taxa of phytoplankton.

In the field, as we explain, the measured growth rates were ***net growth rates*** as there was active grazing i.e. field growth rates includes losses from grazing. This means that the 'high' growth rates observed in the field were not directly comparable to those in the MART, where these was no grazing. It was expected that the measured field growth rates would be lower due to grazing masking the true growth rate. Actual growth rates in the field were likely to be more comparable to the those in the MART experiments. Morison et al. (2019) measured grazing rates on the same cruise, so we can get an idea of how grazing impacted growth rates (see lines 636-639 in the discussion); grazing rates were between 0.26 to 0.44 day$^{-1}$, so true growth rates were the observed net increase in phytoplankton (determined from flow cytometry or chlorophyll) plus the

component lost due to grazing. Our net growth measurements were based on direct measurements of changes in the water; Morison et al.'s (2019) grazing rates was based on on-deck experiments, so it would be inappropriate to add a correction by pairing specific grazing rates to our growth measurements – but the results of Morison do give us an approximate estimate of grazing.

A difference between our work and previous MART and mesocosm experiments is that we know there were no grazers in our simple MART experiments. Experiments using filtered seawater (e.g. passed through a 50 µm screen in Wang et al. 2015) will remove large zooplankton, but not smaller grazers (e.g. heterotrophic protists) from the water. This will lead to selective grazing on smaller phytoplankton in many of the previous MART or mesocosm experiments and an unnatural community structure.

**Changes to the text:**

For clarification regarding physiological status, we changed the first lines of section 2.3.2 to read: "Variable chlorophyll fluorescence was used as an indicator of physiological status of the phytoplankton in the MART. The maximum quantum yield of photosystem II ($\phi_{PSII}$) was measured using the saturating pulse method (Genty et al., 1989; Maxwell and Johnson, 2000). This parameter measures the proportion of light absorbed by the photosynthetic pigment chlorophyll in photosystem II that is used to drive photochemistry (Maxwell and Johnson 2000)."

The reviewer is correct to note that there was a second 'spurt' of growth in the experiment with *Synchococcus* (Fig. 2), though this was not as significant as suggested by the increase in the quantum yield of photosystem II (which, as discussed above, is not an indicator of biomass and therefore cannot be used to calculate growth rates). The second spurt of growth was always acknowledged in the results section (lines 336 to 338):

"There was a second period of growth during which the biomass of *Synechococcus* increased from mean values of 15.3 to 26.0 µg chl. *a* L$^{-1}$, but the growth rate was relatively low (0.21 day$^{-1}$) compared with the initial period of relatively fast growth (1.34 day$^{-1}$)."

In response to the reviewer's concerns, the following text has been added to the discussion (lines 590-600:

"After the initial phase of rapid growth during the *Synechococcus* MART experiment there was a decline in biomass, as indicated by the halving of chl. *a* concentration between days 4 and 8. This was followed by a second period of growth, which did not correlate with relatively warm ice nucleation (Fig. 2). There are two possible explanations for this discrepancy. Firstly, the second period of growth (0.21 day$^{-1}$) was not rapid compared with growth rates during the early phase of the culture (1.34 day$^{-1}$). Secondly, the composition of the water had changed significantly since the initial phase of fast growth due to the death and physiological stress of

*Synechococcus* between days 4 and 8. In addition to the declines in chl. *a* concentration and quantum yield of photosystem II, there was also an increase in the amount of dissolved organic matter in the MART, as indicated by an increase in FDOM concentration between measurements on days 1 and 6 (Fig. S3). While MART and mesocosm experiments often progress with one distinct peak (such as in the experiment with *Thalassiosira weissflogii* (Fig. 3)) and compilation of growth curves in Lee et al. (2015), there are examples where there is more than one peak in phytoplankton biomass over the course of an experiment (Lee et al., 2015; McCluskey et al., 2017; Wang et al. 2015)."

**Reviewer's Comment: Aerosol measurements:** The "estimated" total aerosol concentrations (Figure S1) need to explained in the methods section. The PAS measures optical diameter, whereas CPC measures total particles above some lower diameter threshold. What values were used to convert optical diameters to volume equivalent or geometric diameters? How many days were both the PAS and CPC available for generating the relationship between total particle concentrations and PAS particle concentrations? There is an $R^2$ value (0.7), but no data is shown and based on Figure S1 there were no data points that represented the highest "estimated" concentrations. Given the amount of variability and that variability in aerosol amount will be a major driving component of the aerosolized INP collected during the experiment, this is a vital detail of this experiment that is not adequately measured or described.

**Authors' response:** The aerosol measurements are first discussed in lines 127-134 and Figure 1 of the original text, "Total aerosol concentration (in the approximate range of ~ 0.01 to ~1 µm diameter) was measured with a water-based Condensation Particle Counter (CPC, TSI, Inc. Model 3786). In addition, a Portable Aerosol Spectrometer (GRIMM 1.108) was used to measure the size-resolved number concentration of 0.3 to 20 µm diameter aerosol."

And aerosols are again discussed in lines 431-440 which stated "Information on additional variables which were assessed, but did not indicate a clear relationship to ice nucleation behavior is provided in the Supplement. Aerosol number concentrations in the MART were greater in the headspace over cultures (Fig. S1) compared with the headspace over blank seawater (Table S1), indicating that biomass and biological activity affected the production of aerosol. Aerosol number concentration measured by the CPC was generally higher (mean = 2.21 x $10^6$ L$^{-1}$) over the culture of *T. weissflogii* compared with *Synechococcus* (mean = 1.43 x $10^6$ L$^{-1}$), reflecting the relatively higher biomass in the diatom culture, as indicated by chlorophyll *a* concentration in Figs. 2b and 3b above. Less aerosol were generated from a control MART tank containing only artificial seawater (< 1 x $10^6$ L$^{-1}$; Table S1). Aerosol number concentration increased in the control experiment after the addition of L1 nutrients, possibly to the addition of organic matter in the form of vitamins, and the growth of bacteria in the water (Table S1). In summary, there is no connection between aerosol concentration and ice nucleation activity indicating that the improvements in ice nucleating ability are not driven by the total aerosol available to catalyze freezing."

Given the MANY measured variables included in this study, (note the 5 panel figures!) and the fact that the aerosol-INP results are underwhelming in that no correlations were found as reported in the text, we think this discussion of aerosols is adequate for the main text. Thus, we prefer to keep the aerosol data in the supplement.

Now, the Reviewer calls into question our estimates of total aerosol concentration from the PAS optical sizing instrument, for periods when the better instrument for measuring total aerosol concentration (the CPC) failed. As the Reviewer is probably aware, the 2 instruments measure different size ranges, so this estimate relies on the underlying assumption that the size distribution doesn't shift from day to day. Also the techniques operate on different physical and optical methods so the uncertainty in the estimate has to be pretty high. On the plus side, this estimate was only used for 1 day that coincided with ice nucleation data, Day 7 in the mesocosm with *Thalassiosira weissflogii*. And let us also be specific that it only impacts Figure 3B, in which the concentration of INPs is reported. All the aerosol data used to calculate INP concentrations on other days is measured directly from the CPC.

Based on this comment, we thank the reviewer for questioning this estimation, which causes us to realize it is stronger to sacrifice the estimated data (1 day) and keep the measured data. Based on this comment, we have removed the Day 7 INP concentration data from Figure 3B. In the supplemental, the aerosol data is now modified to include just the measured CPC data. We note that CPCs are often used to measure "total" aerosol, but in reality they measure ~ 0.01 to ~1 micrometer diameter aerosols, whereas PAS measurements are size-resolved measurements between 0.3 and 20 micrometers. Given that aerosol collected on the L1 stage of the PIXE impactor (corresponding 0.06 to 1 micrometer aerodynamic diameter) were used for ice nucleation measurements, the CPC data were the most appropriate for the estimation of INP number concentrations as they align with the size range of the CPC.

**Reviewer's Comment: Overgeneralization:** This paper includes too much general language describing "relatively high" and "relatively warm temperature INP" that is misleading. As described above, the INP number concentrations are very low compared to any marine INP data in the literature. Additionally, the ice nucleation community tends to reserve "warm temperature INPs" terminology for INPs that are higher than -15C or even -10C, whereas all but one sample (that nucleation at -15C) nucleated at temperatures lower than -18C. "Aerosol sampled over phytoplankton cultures grown in a marine aerosol reference tank (MART) induced nucleation and freezing at temperatures as high as -15.0 ∘C during exponential phytoplankton growth." – Please note that only one sample (collected on day 3 of the first experiment) froze before -19C.

**Authors' response:** We agree that there is no formal terminology for categorizing aerosols according to their ice nucleation ability, This is why we use the word relative. As discussed in our response above, we have improved the section on comparisons to previous studies in the literature, both with regards to INP concentration and to temperatures. The more specific details in that section alleviate the overgeneralization issue raised here.

**Specific Comments:**

**Reviewer's Comment:** Introduction:

The authors should include a statement on the relative role of marine INPs compared to other INP sources; That is, sea spray aerosol is a weak source of INPs compared to dust (e.g., DeMott et al., 2016) and marine INPs are likely only relevant in remote regions (Zhao et al., 2021)

**Authors' response:** See our response to the main comment on INP above. In addition, the following has been added to the introduction (Lines 57 to 62):

"Number concentrations of INP from marine sources are poorly constrained, but generally considered to be orders of magnitude lower than number concentrations of INP from terrestrial sources, particularly mineral dust (Beall et al. 2022; DeMott et al., 2016). Observations indicate that number concentrations of INP over the remote ocean are low (0.38 to 4.6 IN m$^{-3}$ of air over the Southern Ocean nucleating at -20 ºC) (McCluskey et al. 2018). Nevertheless, marine sources of INP are important over the remote oceanic areas, such as the Southern Ocean, due to the absence of terrestrial INP and large surface area of the Earth covered by the ocean (Zhao et al., 2021)."

**Reviewer's Comment:** Methods:L99 – please describe the order of events for the nutrient addition in the two phytoplankton culture experiments – what day were the nutrients added?

**Authors' response:** The following text has been added (lines 111-113):

"Nutrients (nitrate, phosphate, silicate, trace metals, and vitamins) were added on the morning of the phytoplankton addition to the tank. The nutrients were mixed into the seawater using the stirrers (Fig. 1) for approximately an hour before the phytoplankton were added."

**Reviewer's Comment:** L123 – What is the mixing tube that is included in Figure 1? It is not explained in the text.

**Authors' response:** The text (lines 129-130) now reads: "The air was dried and passed through a 46 cm long glass mixing tube, with a 0.64 cm I.D. entrance ports on one end, and 0.64 cm I.D exit points on the opposite end."

**Reviewer's Comment:** Figure 1 – Where is the purge line that is described in Section 2.1 (L110)? Also, please add the flow rates for the various instruments, as currently it looks very possible for the CPC, PAS, and PIXIE impacts to have pull air from the exhaust. Is there a constant filtered input flow to avoid negative pressure building in the tank? The "Exhaust" included in this diagram is not described.

**Authors' response:** We performed routine tests on the flow rates throughout the system as well as the exhaust, as is common practice in aerosol measurements. The exhaust is mentioned in the Figure 1, in line with the level of detail in numerous other manuscripts in the literature (e.g. Lee

et al., 2015, Wang et al., 2015). Figure 1 is a schematic representation of the experimental set up and we do not feel that it is necessary to add more detail and potential confusion to the diagram.

**Reviewer's Comment:** Flow rates and particle values during non-plunging periods would be helpful to demonstrate the experiment was conducted successfully.

**Authors' response:** Again, this suggestion is frankly, unnecessary.   In our opinion, cluttering the figure with a list of flow rates would not provide a helpful visual. Also, we wonder what the Reviewer means by "particle values during non-plunging periods" but we are satisfied with the figure without them.

In response, we now add to the text (lines 132-134): "At the beginning of each sampling period, a Gilbrator flowmeter was used to confirm a positive flow outwards through the HEPA exhaust filter."

**Reviewer's Comment:** L280 – Nutrients (L1), growth medium, a "exponentially growing culture", and silicate were added to the ASW for the culture experiments (Section 2.1); the control ASW experiment only accounts for nutrients. What are the potential influences of these additional components (growth medium and silicate) potential influences on INPs? Given that INPs are rare in the atmosphere, but difficult to avoid in laboratory experiments, this is a major concern, as the initial signal may very well be a signal from an initial emission of INPs from the medium used that fell off after a few days. In fact, a study published by Fröhlich-Nowoisky et al. (2015) found that the initial medium choice was ice nucleation active at temperatures lower than -12C ("We originally intended to grow the isolates on malt extract agar. However, since the available product was found to contain some IN (active at $-12$ ∘C), an approximate equivalent using IN-free ingredients (tested to $-18$ ∘C) was constructed.")

**Authors' response:** This has been addressed under the 'Experimental INP background' above. On the specific issue of silicate – silicate is an essential nutrient for diatoms, so it was necessary to add silicate to the *Thalassiosira weissflogii* culture. Silicate is not necessary for cyanobacteria, but we added it to the *Synechococcus* culture for consistency. Note that silicate has been added to other experiments of this type at concentrations of 10.6 and 106 µM (Lee et al., 2015; Wang et al., 2015), compared to 60 µM $Na_2SiO_3$ in our experiment. Measurements of the procedural blanks from ASW alone did not include silicate, measurements of the ASW+L1 blank included silicate (and the other nutrients). By looking at the difference between the two procedural blanks, we can see the effects of the nutrients. The mean ice nucleation temperature of the procedural blank containing only ASW was -34.9 ± 0.9 ºC, compared with -31.3 ± 0.9 ºC for ASW+L1 nutrients. This indicates that the nutrients had direct or indirect effect on ice nucleation (as discussed in the text).

**Reviewer's Comment:** The "estimated" total aerosol concentrations need to explained in the methods section. The PAS measures optical diameter, whereas CPC measures total particles above some lower diameter threshold. What values were used to convert optical diameters to volume equivalent or geometric diameters? How many days were both the PAS and CPC available for generating the relationship between total particle concentrations and PAS particle concentrations? There is an $R^2$ value (0.7), but no data is shown and based on Figure S1 there

were no data points that represented the highest concentrations. Given the amount of variability and that this is a driving component of the aerosolized INP collected during the experiment, this is a vital detail of this experiment.

**Authors' response:** See response above.

**Reviewer's Comment:** ResultsFigure 2a – what is plotted for freezing temperature? Is this temperature 50% frozen? The values do not correspond to Figure 2c. E.g., Day 5 in Figure 2c has the first non-zero value at a temperature greater than -30C (looks like -29C), but the value plotted on 2a is lower than -30C.

**Authors' response:** Figure 2a shows the mean (average) freezing temperature on each day (mean ± pooled SD). This is stated in the figure legend.

**Reviewer's Comment:** Figure 2c – is this the average fraction frozen from the 25 repeat analyses? If so, please indicate here or clarify what was done in the methods section.

**Authors' response:** Yes, this is the mean freezing temperature from 25 repeated freeze-thaw cycles. The text has been changed in the Methods to (lines 171-173): "Mean ice nucleation temperature and fraction frozen over the temperature range (0 to -40 °C) were determined from multiple ice nucleation events observed on the same sample by repeating the cooling and warming cycle 25 times. In practice, some runs resulted in a data set with fewer than 25 freezing points due to droplet evaporation."

**Reviewer's Comment:** Figure 3a – same as comment above for Figure 2a.

Figure 3a shows the mean (average) freezing temperature on each day (mean ± pooled SD). This is stated in the figure legend.

**Reviewer's Comment:** L365 – "High values of PSII were associated with ice nucleation at relatively warm temperatures in both taxa." – I do not see this association in Figure 2b. Why is there not an increase in INPs after Day 10?

Line 400 to 401 has been changed to: "High values of $\varphi_{PSII}$ (> 0.44) were associated with ice nucleation at warmer temperatures (> -24 ºC) during the *T. weissflogii* experiment."

**Reviewer's Comment:** L391 – "The maximum concentration of INPs in both experiments was approximately 2 x 10-3 INP L-1 air." - what temperature?

**Authors' response:** The sentence (lines 428-430) has been changed: "The maximum concentration of INPs in both experiments was approximately 2 x $10^{-3}$ INP L$^{-1}$ air at -26 ºC (*T. weissflogii*) and -29 ºC (*S. elongatus*)."

**Reviewer's Comment:** L392 (and throughout) – "Relatively high concentrations of INPs at warmer temperatures were observed during exponential growth on days 2 and 3 in both T. weissflogii and S. elongatus." – There are many instances where the authors stat "relatively high

concentrations", when the INP number concentrations are not really compared to anything. Note that INP number concentrations over marine regions are usually greater than $0.001 L^{-1}$ at -20C from mesocosm experiments (McCluskey et al., 2017, McCluskey et al., 2018, Mitts et al., 2021), aerosol sampled over the north Atlantic (McCluskey et al., 2018, ), aerosol sampled over the Southern Ocean (McFarquhar et al., 2021), or any open ocean region (Welti et al., 2020).

**Authors' response:** This sentence has been deleted (line 392). Number concentrations of INP have been discussed above under '**INP number concentrations**'

**Reviewer's Comment:** L398 – "Aerosol number concentration was generally higher (mean = 2.21 x 106 L-1) over the culture of *T. weissflogii* compared with *S. elongatus* (mean = 1.43 x $10^6$ L-1), reflecting the relatively higher biomass in the diatom culture, as indicated by chlorophyll a concentration in Figs. 2b and 3b above." – How is the biomass and diatom culture in the seawater influence aerosol number?

**Authors' response:** We too, are interested in what factors drive the relationship between biomass in the ocean and aerosol number, but our study doesn't provide answers to that question.

**Reviewer's Comment:** Figure S1 – There is a lot of variability in particle number concentrations. It would be informative to also share the aerosol size distributions from the MART and see how those change. Note that the authors need to include the size range that these aerosol concentrations are for (are they total or just for particles with diameters of specific sizes?

**Authors' response:** The CPC is our main aerosol instrument here. CPCs are often considered a measure of total aerosol, though in fact our CPC samples only particles up (in the approximate range of ~ 0.01 to ~1 micron diameter). This aligns well with the size range of aerosols collected onto the PIXE impaction stage for the ice nucleation measurement. Also, more generally, designating the CPCs as the total aerosol measure is reasonable for most natural applications since the vast majority of atmospheric particles are in the submicron range. However, in a marine environment, there may also be significant supermicrn aerosol.

**Reviewer's Comment:** L550 – "Bloom collapse has been proposed as a major source of organic matter in SSA (O'Dowd et al., 2015) and INPs were associated with the collapse and decay of phytoplankton blooms during mesocosm experiments with natural seawater (McCluskey et al., 2017)." – Please specify that the McCluskey et al. (2017) study argued that the increase in INPs was associated with a particular temperature range ("increases in INPs active between -25 and -15C lagged the peak in Chl a in both studies, suggesting a consistent population of INPs associated with the collapse of phytoplankton blooms."

**Authors' response:** The text has been changed to (lines 669-702):

"Bloom collapse has been proposed as a major source of organic matter in SSA (O'Dowd et al., 2015). McCluskey et al., (2017) found an increase in the production of INPs, active between -25 and -15 ºC, following the peak in chlorophyll concentration in a mesocosm experiment with natural seawater. They explained this consistent population of INPs as resulting from the

collapse and decay of the phytoplankton blooms during the mesocosm experiments (McCluskey et al., 2017)."

[revised manuscript text omitted]

---

## Author Response (AR2)

**Production of aerosol containing ice nucleating particles (INPs) by fast growing phytoplankton**

Daniel C. O. Thornton[1], Sarah D. Brooks[2], Elise K. Wilbourn[1], Jessica Mirrielees[2], Alyssa N. Alsante[1], Gerardo Gold-Bouchot[1], Andrew Whitesell[1,3], Kiana McFadden[2,4]

**Public justification (visible to the public if the article is accepted and published)**:
I would like to thank the authors for incorporating the suggestions made by both reviewers. The revised version looks pretty good and it is almost ready for publication; however, there are some minor comments that need to be properly addressed before I can accept the manuscript.

**Reviewer #1:**
The conclusion could benefit from a short introduction sentence e.g. "In this study we investigated … "., since the length of the manuscript increased within the review process. The authors could bring back the reader to the overall big picture and summarize and conclude their results better.

Following the reviewer's suggestion, we have added the following (Lines 702-703):

"In this study, we investigated whether the physiological status of phytoplankton, especially growth rates, can be linked to the properties of primary marine aerosol and ability to act as effective INPs."

Clearly, the high freezing temperature of the procedure blank is the main issue of the presented methodology, limiting the interpretation of the results as I already mentioned in my first comment. Although, the authors already improved the methodology between the two sampling campaigns, further improvements could be considered for future studies. I whish that the authors could include a (short) outlook in the conclusion that helps designing further campaigns. The authors can think of experimental improvements that allow to lower the procedure blank freezing temperatures and share it with the scientific community to help build upon their research.

As the reviewer noted, we have already improved the methodology in the previous revision. We agree that blanks are an issue for offline ice nucleation measurements in general. This important topic could be the topic of a future review paper, but we do not feel that the conclusion of this paper is the correct place for such a discussion.

Line 102: Change "The ASW was made with high purity analytical grade salts. Nevertheless, the large mass of salts in artificial seawater represents a source of potential contamination" to "The ASW was made with high purity analytical grade salts. Nevertheless, the large mass of salts in artificial seawater represents a source of potential contamination for ice nucleation experiments".
Changed as suggested.

Line 165: sometimes you write "dewpoint" and sometimes "dew point".
Changed to 'dew point' throughout the manuscript.

Line 253: "The image analysis method was based on Engel (2009), see Thornton and Chen (2017) for details." Can you include an example image in the SI?
Figure S2 has been added to the Supplementary Information – this shows unprocessed light microscopy images (i.e. the 'raw data') for TEP and CSP samples collected from the MART during the growth of *Thalassiosira weissflogii.*

Line 541: Change "… GRH hypothesis … " to "… GRH … " since the acronym already implies the word hypothesis.
Corrected as suggested.

Line 684: Change "Different phytoplankton have different have significantly different …" to "Different phytoplankton have significantly different …"
Corrected as suggested.

**Reviewer #2:**
L331: "… per day and mean INP freezing temperatures…" – can this be added to the figure 2a caption? Right now it only refers to the temperatures as "INP freezing temperatures" rather than "mean INP freezing temperatures". There are inconsistencies throughout and perhaps the same freezing temperature is referred to throughout the text, in which case it may be simplest to state in the methods "mean freezing temperatures are used to described INP spectra".
Changes have been made to the Results text and figure legends (Figures 2 and 3) to make it clear when temperatures represent 'mean freezing temperatures'.

L 338: "freezing temperatures" should be "mean freezing temperatures"?
Changed as suggested.

Figure 3a: are these mean freezing temperatures, or onset freezing temperatures (L385-386)? Can this also be added to the figure caption (specify if the plotted temperatures in 3a are mean, median, or onset)?
Changes have been made to the Results text and figure legends (Figures 2 and 3) to make it clear when temperatures represent 'mean freezing temperatures'.

L388: "Fraction frozen curves showed that freezing occurred at temperatures significantly warmer than the procedural blanks at the beginning of the experiment (days 2 and 3) and at the end of the experiment (days 16 and 29)." – What about Day 7, which is above the ASW+L1 data for many temperatures? I also think it's important to point out that days 2 and 3 are >7? degrees higher than the procedural blanks and days 7, 16, and 29 are only higher by ~2 degrees (or whatever these specific values are).

The explanation of what's happening in this graph is already included in the previous line of the existing text (lines 391 to 392):

"The onset of ice nucleation on days 2 and 3 were -19.1 ºC and -22.3 ºC, respectively (Fig. 3c). On these days, the fraction of INPs frozen reached 100% at -24 ºC (Fig. 3c). In contrast, the

onset of nucleation was < -27 ºC for sampling days at slower growth rates later in the experiment (Fig. 3c). Fraction frozen curves showed that freezing occurred at temperatures significantly warmer than the procedural blanks at the beginning of the experiment (days 2 and 3) and at the end of the experiment (days 16 and 29)."

We have not added any text describing Day 7 as we hope that the reader can discern sufficient information from Figure 3C itself. It would extend the results section significantly if we were to describe all the individual plots.

L705: "Significantly, our results are the first to show that fast growing phytoplankton are a source of INPs that catalyse freezing at relatively warm temperatures" – What temperatures, specifically?

The following text has been added (lines 714 to 716):

"Mean freezing temperatures during the early growth phase of the MART cultures were > -24 ºC, which was warmer than the mean freezing temperature of the procedural blanks (-34.9 for ASW and -31.3 ºC for ASW+L1 nutrients)."

**Editor:**
L44 and L61: There is a missing space after the "period".
Corrected as suggested.

L73: I think "limitation, stressors that are" should be "limitation and stressors that are"
Changed to the following (lines 72 to 73):
"The amount of DOM released by phytoplankton increases when cells are stressed by environmental factors associated with bloom collapse, such as nutrient limitation (Thornton, 2002; 2014).

L98: Here and a long the text "h" and "hour" are used. I suggest using "h" along the text.
'hour' has been replaced with 'h' in several places, particularly in the methods.

L100: "L1": please define it, especially because a similar term is used for the PIXE cascade impactor. Actually, I suggest changing the nomenclature of the PIXE stages to avoid confusions.
Lines 99 to 100 have been changed to the following:
"Prior to introduction to the MART, phytoplankton were grown in artificial seawater (ASW) (Harrison et al., 1980; Berges et al., 2001) supplemented with trace metals and vitamins from the L1 medium recipe of Guillard and Hargraves (1993)."

L109: Here "Thalassiosira weissflogii" is called "T. weissflogii"; however, in the following text the authors jump back and forth between both names. Please after L109 call "Thalassiosira weissflogii" as "T. weissflogii" (including figures and tables).

*T. weissflogii* has been used throughout the text after its initial introduction by its full scientific name.

L136: "Samples collected on L1 impaction stage (0.06-1 µm aerodynamic diameter)". Based on the previous line, it seems that L1 contains the larger particles. Please double check this.

We thank the Editor for catching this. The stages were listed out of order (6, 3, and L1). This has been corrected.

The text now reads: "… PIXE cascade impactor with the following stages: 6, 3, and 01, corresponding to 8, 1 and 0.06 µm diameter, respectively (Fig. 1). Samples collected on the L1 impaction stage (0.06-1 µm aerodynamic diameter) were analysed. Aerosol were collected for 2 h at an air flow of 1 L min$^{-1}$ through the sampler. Samples were stored at -80 °C."

L199 and the following text: "ml" should be "mL".
Changed as requested.

L258: "(EEm)" is not necessary as it was not used in the following text.
The acronym 'EEM' has been deleted.

L331 and the following text: "Synechococcus elongatus" is called "Synechococcus". Please use the entire name (or S. elongatus if appropriate).
*S. elongatus* has been used throughout the text to refer to the organism grown in the MART. It is listed as *S. elongatus* in the culture collection it was obtained from. The term *Synechococcus* is used to refer to organisms counted in the field by flow cytometry as we could not determine which 'species' or strains were encountered in the North Atlantic. The use of *Synechococcus* is consistent with Wilbourn et al. (2020), which originally presented the field data that was used to calculate growth rates in this manuscript. Further, it is common in the biological oceanography literature to use the genus name *Synechococcus* without a species name as the genetic diversity of this organism does not fit into conventional definitions of species.

L403: DOM was defined in L70
Corrected.

L442: TEP and CSP were defined in L236
Corrected.

L442: FDOM was defined in L256
Corrected.

Figure 1: What is the meaning of PIXE A and PIXE B? in L135, only 1 cascade impactor with 3 stages is mentioned.
This has been explained in the revised text on sample collection with the PIXE impactors (lines 135-140):

"For offline ice nucleation measurements, size-sorted aerosol samples were collected on combusted aluminium foil substrates inside a PIXE cascade impactor with the following stages: 6, 3, and 01, corresponding to 8, 1 and 0.06 µm diameter, respectively (Fig. 1). Samples

collected on the L1 impaction stage (0.06-1 µm aerodynamic diameter) were analysed. Aerosol were collected for 2 h at an air flow of 1 L min$^{-1}$ through the sampler. Samples were stored at -80 °C. There were two PIXE cascade impactors in the system (Fig. 1), but only samples from PIXE A were used in the analysis. PIXE B served as a backup system to ensure that back-up samples were available if there were issues such as instrument failure."

Figure 2: Please include the meaning of the open circles and green squares as in Figure 3.
The Figure 2 legend has been corrected and is consistent with Figure 3.

Figure 4. I think the y-axis in both panels can go in logarithmic scale.
We agree that it is convention to plot these type of data on a log scale. This convention arises from the wide range of concentrations of INP observed in the atmosphere at different temperatures. Data collected during one field campaign can cover several orders of magnitude in terms of INP number concentration. However, in our MART experiments we observed a narrow range of INP concentrations, both with temperature and across different days of the experiment. Therefore, there was no advantage to plotting our data on a log scale.

Figure 5: I think "Fraction of ice nucleating particles (INPs)" should be "Fraction of droplets"
Yes – this has been changed to 'fraction of droplets' in the figure legend.

Table S1: "bank" should be "blank"
Changed to 'blank'.

Kianna McFadden (co-author) wanted to associate her ORCID digital identifier (0000-0003-1383-2503) with her name, but I could not work out how to add it at this stage of the submission.